# Penalties for industrial accidents: The impact of the Deepwater Horizon accident on BP's reputation and stock market returns

William McGuire[1], Ellen Alexandra Holtmaat [2]*, Aseem Prakash [3]

**1** University of Washington, Tacoma, Tacoma, WA, United States of America, **2** London School of Economics, London, United Kingdom, **3** University of Washington, Seattle, WA, United States of America

\* e.a.holtmaat@lse.ac.uk

**Data Availability Statement:** All data files are available from the Figshare database (https://doi.org/10.6084/m9.figshare.19758640.v1).

## Abstract

Do visible industrial accidents damage firms' reputations and depress their stock market returns, and do these penalties spill over to other firms in the industry? On April 20, 2010, the Deepwater Horizon offshore oil rig in the Gulf of Mexico leased by BP exploded and sank, causing 11 deaths and the largest marine oil spill in US history. We examine the impact of this accident on BP's reputation and stock market performance using data from YouGov's BrandIndex and Capital IQ's financial data for the period 2007–2017. We employ a synthetic control analysis to examine the extent and duration of these penalties. We find that in the aftermath of the Deepwater accident, BP's reputation declined by approximately 50% relative to the synthetic control, and this decline persisted through the end of 2017. Yet, in terms of financial market returns, though the stock price dropped drastically in the first two months, we do not find a statistically significant decline in the stock market returns either in the mid-term (1–2 years) or the long term (2–7 years). In terms of spillover effects, we find no evidence of reputational damage or a decline in stock market returns for other oil and gas firms. These findings suggest that while environmental accidents invite swift and lasting reputational penalties, they might not depress the stock market performance in the long run. Moreover, the impact either on reputation or stock market returns does not necessarily spill over to other firms in the same industry.

## Introduction

Climate concerns are leading to intense scrutiny of fossil fuel firms, including how they access capital, what sorts of subsidies they get from the government, and the ecological damage their activities pose to local communities. "Naming and shaming" of fossil fuel firms, along with their supply infrastructure such as banks and cloud computing companies such as Amazon, is an integral part of the advocacy tool kit of climate groups. While the ecological impact of fracking continued to draw public scrutiny, the issue of offshore oil spills seems to gather less attention [1]. This is problematic because offshore production accounts for about 30% of global crude output, and [2] oil spills occur quite regularly [2].

**Funding:** We like to thank the Swiss National Science Foundation (snf.ch) for the Doc.Mobility grant P1GEP1_181399 for EAH. The funders had no role in study design, data collection and analysis, decision to publish, or preparation of the manuscript.

**Competing interests:** The authors have declared that no competing interests exist.

Industrial accidents can inflict lasting reputational and financial damage on firms. The 1979 Three Miles accident brought a halt to the construction of new nuclear plants in the United States. In the wake of the 2011 Fukushima accident, Germany decided to phase out its existing nuclear plants. Thus, by strategically leveraging both the reputational and financial harms of offshore accidents, the climate movement can exert additional pressure on fossil fuel firms to forgo offshore oil drilling and production, and the financial institutions not to fund such activities.

The 1989 Exxon Valdez disaster undermined the offshore industry's claim about their state-of-the-art safe transportation technologies. In addition to lawsuits seeking financial damages from Exxon, this incident motivated the enactment of the Oil Pollution Act of 1990, which substantially increased oil transportation costs by requiring a double hull design for new tankers and barges. The industry also developed elaborate safety protocols [3].

The offshore oil industry suffered a major disaster in 2010 with the explosion and eventual sinking of the Deepwater Horizon oil platform in the Gulf of Mexico. In this paper we examine how this accident affected the corporate reputation and the stock market return of Deepwater's operating firm, BP (formerly, British Petroleum). We focus on BP while recognizing that its contractors such as Schlumberger, Halliburton, Transocean, and Weatherford were also implicated in the spill [4].

Corporate reputation can be viewed as "the accumulated impressions that stakeholders form of the firm" [5]. It is a crucial asset for any firm [6] because outside stakeholders often do not possess information about the internal workings of the firm. To pass an evaluative judgment on the firm, they rely on proxies such as its corporate reputation. Consequently, firms view corporate reputations as important strategic resources [7] that create benefits such as a better relationship with regulators [8], lower cost of capital [9, 10], and attract and retain customers and managerial talent [11]. Given the substantial payoffs of a good reputation, firms invest vast sums in building and protecting their reputation, be it for high-quality products, ethical conduct, environmental stewardship, or community engagement [12].

But corporate reputations could be fragile. They could be damaged when firms suffer industrial accidents, recall faulty products, or face media scrutiny for poor labor or environmental practices. The Deepwater Horizon accident is probably one such event. On April 20, 2010, a blowout of the Deepwater Horizon offshore drilling rig led to the death of eleven workers and caused the largest marine oil spill in history. The oil spill caused catastrophic environmental damage to ecosystems in the Gulf of Mexico. BP was criticized sharply for the failures that led up to the accident and ultimately pled guilty to eleven counts of manslaughter and other misdemeanor and felony charges. To date, BP has paid over $60 billion to settle criminal and civil complaints along with other fines. We recognize that the reputational impact of industrial accidents is mediated by how the media covers it and the activists frame it [13]. In the case of BP the shock value of the event, the vivid footage of marine pollution, and the media attention were overwhelming [14–16]. Environmental groups also sought to leverage it to promote their goals, including climate change. Thus, the scale of the accident, media coverage, and environmental groups' political mobilization created the expectation of a negative reputational impact.

While BP became a subject of criticism and legal action, it is not clear the extent to which, and for how long, the Deepwater accident affected BP's reputation in the eyes of citizens, the key actors granting firms the "social license to operate" [17]. Firms depend on the external environment for critical resources. They need physical infrastructure to produce goods and services, secure inputs, attract employees, and sell to their customers. For these activities, they rely on a supportive government that provides the appropriate regulatory environment. In addition, firms need a de facto "social license to operate": citizens and communities must view

firms as responsible actors who are meeting societal expectations. Without social legitimacy, firms might find it difficult to access physical inputs and financial capital as well as obtain permits and other resources to function. They may even face political and environmental protests [18]. Of course, profits may help firms to secure social legitimacy. But profits alone may not give them the social license to operate. Hence, even highly profitable firms invest in building their reputations for good citizenship, environmental stewardship, and workplace safety.

In addition to the reputational damage, we examine the impact of the disaster on BP's stock market returns, as each might be driven by different mechanisms. Stock markets are a pillar of the market economy. By providing a summary measure of a firm's performance, the stock price, they play an important role in capital allocation. Moreover, stock prices also provide a measure of the expected future earnings of the firm [19]. Indeed, for many corporations, executive compensation is often linked to the stock price. Thus, without a penalty on stock prices, the long-term implications of the accident for changes in corporate governance (in the absence of new regulations) remain unclear.

Firms' stakeholders typically function in an information-scarce environment. They are also boundedly rational [20] and resort to stereotypes to economize on their limited cognitive capacities [21]. Sometimes, they make assumptions about all firms across the industry based on the actions of a single firm. This leads to the issue of reputational spillovers, where the actions of one firm can bear upon the reputations of other firms in the industry [22]. Indeed, this is an important reason why industry associations often develop industry-wide certification programs to protect the reputation of the industry as a whole [23, 24] and to insulate other firms from any reputation problems that a particular member might face.

Reputational spillovers are not inevitable because stakeholders could differentiate among firms that sell differentiated products. The Volkswagen diesel scandal did not necessarily sully the reputation of Toyota or even other German car companies such as BMW. Nevertheless, reputation spillovers remain a serious concern especially when firms sell an undifferentiated product such as gasoline. We extend our study beyond BP to see if the Deepwater Horizon disaster also affected the reputations of other firms in the oil and gas sector. These include integrated oil and gas companies like Chevron and Shell, exploration and production companies like Marathon, and refining and marketing companies like Valero, Citgo, and Sunoco. Our analysis suggests the absence of any significant reputational spillovers to other firms in the oil and gas industry.

Regarding stock market returns, we find no evidence that the Deepwater Horizon spill diminished BP's stock market returns in the mid (1–2 years) or the long term (2–7 years), despite a drastic drop in BP's stock price immediately after the spill. Moreover, we find no evidence that other firms in the oil and gas industry suffered declines in their stock market returns.

## Data and empirical methodology

A key methodological challenge in assessing the reputational damage or changes in stock market returns from an industrial accident is the absence of a counterfactual: what would BP's reputation have looked like if there would have been no accident? Comparing BP's post-disaster reputation to its pre-disaster one is problematic because it imposes a strong assumption that no other events after April 2010 affected BP's reputation. However, we could compare BP's reputation to that of another firm provided we can establish that the comparison firm's reputation was sufficiently similar to BP's before the disaster. This is the logic motivating the synthetic control method: create a "synthetic brand" that closely resembles BP's reputational record before the accident. We can then compare the change in BP's reputation after the

accident with that of the change in the synthetic brand's reputation for the same period. This approach allows us to establish the impact of the Deepwater Horizon accident on BP's reputation.

BP's Deepwater Horizon accident occurred on April 20, 2010. To study the effect of this accident on BP's reputation, we look for a change in reputation post-2010, with the pre-2010 reputation as the baseline. We draw on data from YouGov's BrandIndex database. These data report respondents' evaluation of the reputations of different corporate brands measured in terms of their general impression of the brand, the perceived quality of the product, the value for money, and the respondent's willingness to work for the company [25]. These data are observed at the brand level rather than at the firm level. This is appropriate for our study because brands are typically the locus of a firm's reputation. The data are recorded daily, but we construct monthly aggregates to make the model estimation less demanding. The data are collected through surveys and used to calculate "scores" by subtracting negative feedback from positive feedback. The scores can range from -100 to +100. A score of zero indicates equal amounts of positive and negative feedback. Scores closer to -100 indicate a predominance of negative feedback, while scores closer to +100 indicate a predominance of positive feedback.

Our objective is to compare the change in BP's reputation before and after the accident to that of a "synthetic control" brand. This brand needs to be sufficiently similar to BP in terms of various attributes and yet should *not* have experienced the negative reputation impact of the Deepwater accident. This approach would reveal how BP's reputation may have fared in the counterfactual scenario where the accident did not take place.

What might this control brand be? Rather than choose the control brand at random or by appealing to theory, we employ a nonparametric estimation method, "synthetic control," to construct a counterfactual brand for BP, based on a weighted average of fifteen other brands (listed in Table 1 below) in our data set. The synthetic control estimator selects brands from our dataset of 660 brands based on their similarity with BP in terms of variables that correlate with corporate reputation (as opposed to the specific measure of corporate reputation that we employ as our dependent variable). These include respondents' reported impression (favorable

**Table 1. Components of synthetic control.**

| Weight | Brand |
|---:|---|
| 0.756 | Shell |
| 0.054 | Craigslist |
| 0.046 | Verizon Wireless |
| 0.027 | Walmart |
| 0.026 | Big Lots |
| 0.012 | Cialis |
| 0.009 | Visa |
| 0.004 | Costco |
| 0.004 | TJMaxx |
| 0.004 | YouTube |
| 0.001 | Red Bull |
| 0.001 | Abercrombie & Fitch |
| 0.001 | Kohl's |
| 0.001 | Sunoco |
| 0.001 | Ikea |

Note: Weights sum to one by construction

or not) of the brand, their sense of the value they get from the brand, the perceived quality of the brand, their satisfaction with the brand, their willingness to recommend the brand, and the "buzz" they've heard associated with the brand. The estimator then selects and attaches weights to these other brands to minimize the difference between selected variables used to construct the synthetic control and BP before "treatment" (i.e., the 2010 accident). The following is the weighted average "control" brand that is most similar to BP before the disaster:

## Reputation effect

The synthetic control method is nonparametric, meaning there is not a simple hypothesis test we can use to judge whether the accident had a "significant" effect on BP's reputation. Instead, we can look at the apparent size of the treatment effect and perform additional robustness tests to ensure our result is meaningful. We begin by examining the standard synthetic firm graph, comparing the outcome for our treated brand (BP) against our synthetic control. This is presented in Fig 1 below:

The vertical line indicates April 2010, the month of the Deepwater Horizon accident. We can see that our "treated unit" (BP) performed quite similarly to our synthetic control before the accident. Both exhibited a long-term upward trend in their reputation scores with a similar short-term variation.

After the 2010 accident, BP's reputation dropped more than 50 points relative to the synthetic control (recall, the reputation scale range is 200 points, from a minimum of -100 to a maximum of +100 points). Though BP's reputation recovered almost half of its losses over the next 18 months, it did not recover to the same level even by December 2017, the end of the

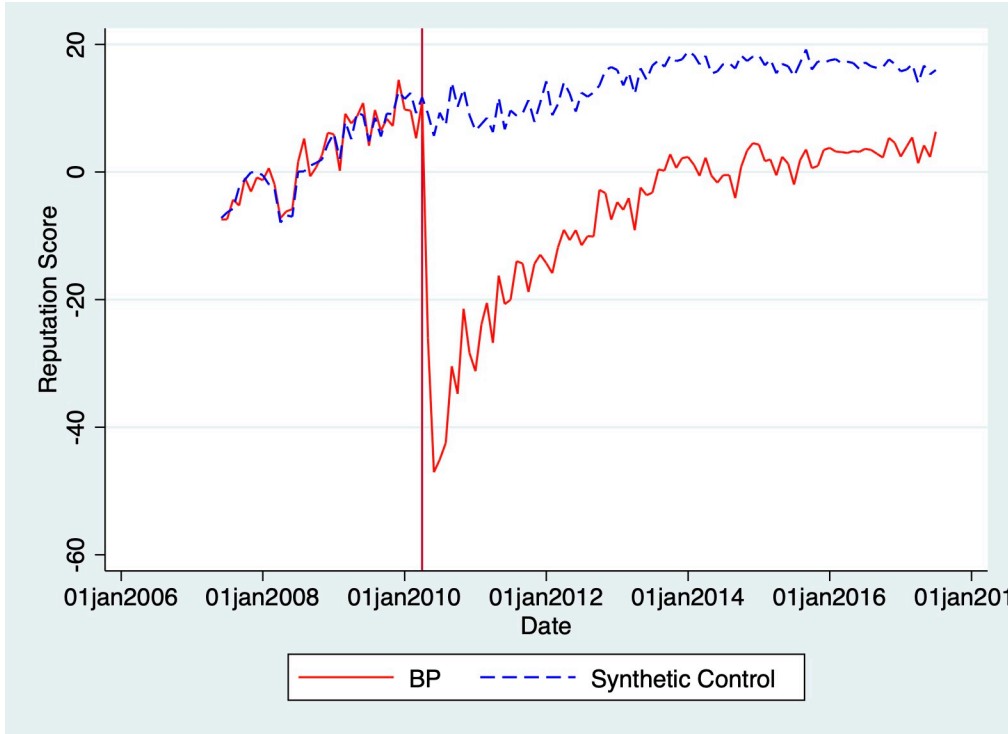

**Fig 1. Reputation of BP vs. synthetic firm.** Note: As Shell–which may itself have been impacted by the oil spill—is an important component of our synthetic control, we form a synthetic control excluding oil and gas firms as well. To address potential concerns that the brands in the synthetic control are not sufficiently similar to BP, we also form a synthetic control based on only oil and gas firms. We plot the synthetic controls in a graph in the S1 Appendix.

period covered by our data set. BP's reputation seems to have stabilized around a lower level (15 points) compared to the synthetic control, suggesting BP has suffered long-term reputational damage from the Deepwater accident.

We can also see the extent of the reputational damage by plotting the gap between the synthetic firm's reputation and BP's. Fig 2 presents this comparison, where a value of zero on the vertical axis represents no difference between the reputation of BP and synthetic control.

This figure has the same intuitive interpretation as Fig 1. We can see the gap widened to approximately 50 points in the months immediately following the accident. About half of the reputational damage was undone within the next 18 months, but the size of the gap decreased only very slowly by the end of the period covered by our data set. Potentially media campaigns by BP to regain public trust might have played a role in undoing some of the initial reputational damage [16].

**Robustness.** Our reputational comparisons are meaningful only if the synthetic control brand is a good estimate of BP's reputation in the counterfactual scenario where the Deepwater Horizon accident did not happen. Following the synthetic control literature, we perform a series of "placebo" tests to substantiate this claim [26]. The objective is to explore whether brands that make up our synthetic control experienced shifts in reputation similar to BP's around the time of the Deepwater Horizon accident. If so, that would call into question our estimate of the reputational shock in Fig 1.

To conduct the placebo tests, we construct synthetic controls for each of the 15 brands listed in Table 1. Thus, for say Walmart, we construct a synthetic control using their own unique set of comparison brands. Our objective is to assess if reputational gaps between the focal brand and its synthetic control are similar to those revealed in Fig 2, where we examined the

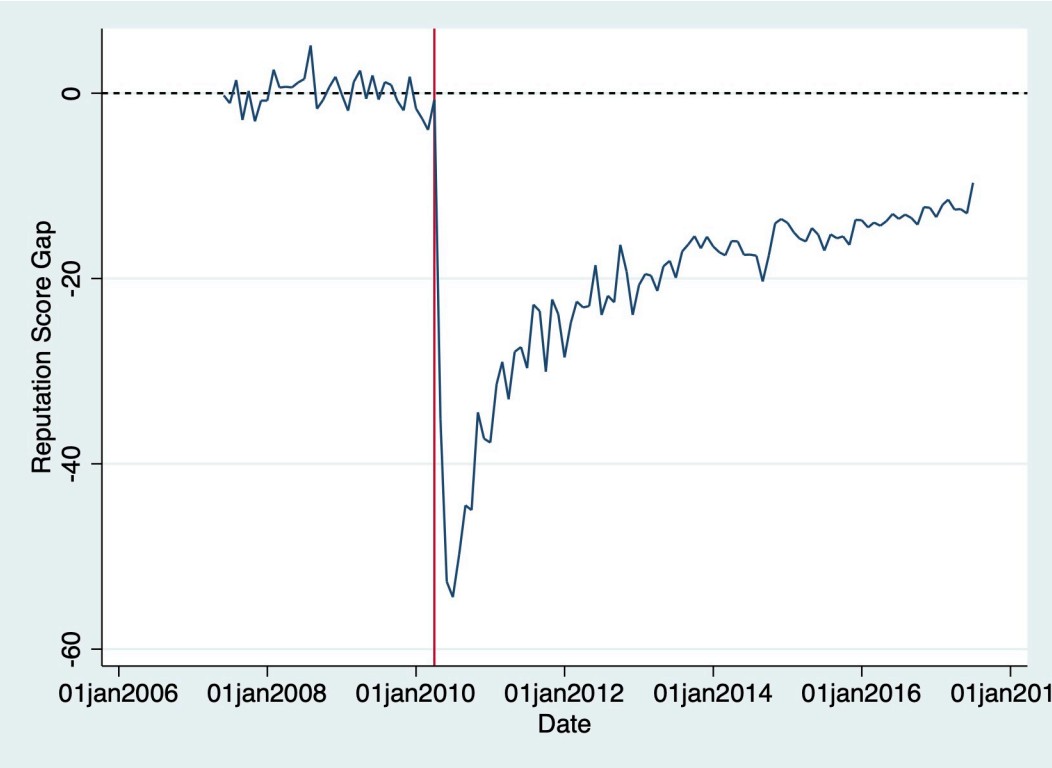

**Fig 2. Gap between BP and synthetic control reputation.**

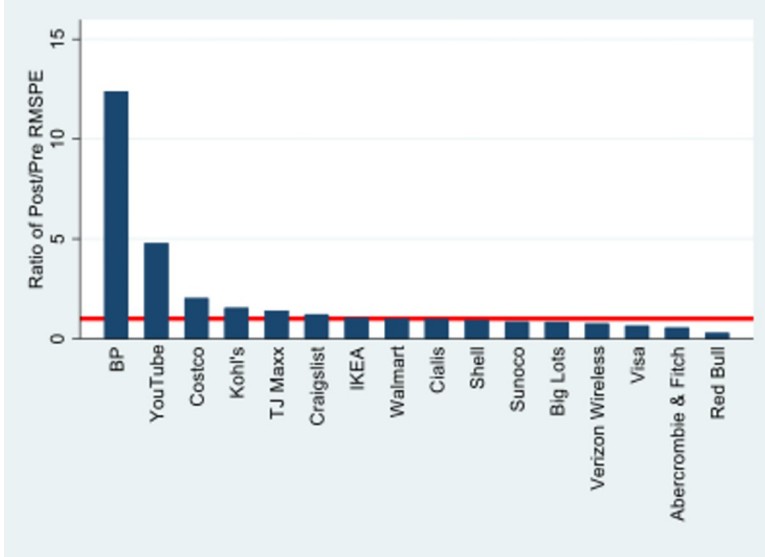

**Fig 3. RMPSE ratios for BP and components of synthetic firm.**

reputational gap between BP and its synthetic control. We do this by calculating the root mean square prediction error (RMSPE) in the case of BP vs. synthetic control and the RMSPE in the case of each brand in Table 1 vs. their respective synthetic controls. The RMSPE measures the extent to which the actual performance of the brand deviates from what we would have predicted based on the performance of the synthetic control brand. When treatment effects are large, we should expect a very low RMSPE before treatment and a large RMSPE after treatment. Fig 3 shows the ratio of post- vs. pre-April 2010 RMPSE for BP and the components of its synthetic firm list in Table 1.

The horizontal line corresponds to a value of one, indicating equivalent pre- and post-Deepwater Horizon RMPSE. The ratio for BP is over 12, indicating that the RMPSE was very low before the disaster and very high afterward. This is consistent with a large treatment effect. The ratio for the rest of the brands ranges from 4.8 for Youtube to 0.29 for Red Bull. The highest RMPSE ratios are for YouTube and Costco, but each of these brands collectively received a weight of 0.008 in the construction of the synthetic control. Their relatively large RMPSE ratios do not undermine our estimate of the treatment effects in Figs 1 and 2.

**Spillover effects.** Arguably, because gasoline is an undifferentiated product, an industrial accident in a single oil firm could have tarnished the reputation of other firms in the industry. It is also important to consider potential spillovers from a methodological perspective; the brands from which we selected BP's synthetic control did include other oil and gas firms; in fact, the single largest component of BP's synthetic control is Shell (75% weight). If Shell was positively or negatively affected by the Deepwater Horizon disaster, it would bias our measurement of the reputational shock in Figs 1 and 2.

To test for potential spillovers within the oil and gas industry, we constructed synthetic controls for every other oil and gas brand (individually) included in the YouGov database. These are Arco, Chevron, Citgo, Gulf, Marathon, Shell, Sunoco, and Valero. To account for possible spillovers within the oil and gas industry, we excluded other oil and gas brands from the construction of these synthetic controls. We then calculated the pre- and post-Deepwater Horizon RMSPE for each of these brands relative to their synthetic controls. The ratios of post- vs. pre-Deepwater Horizon RMSPE are presented in Fig 4:

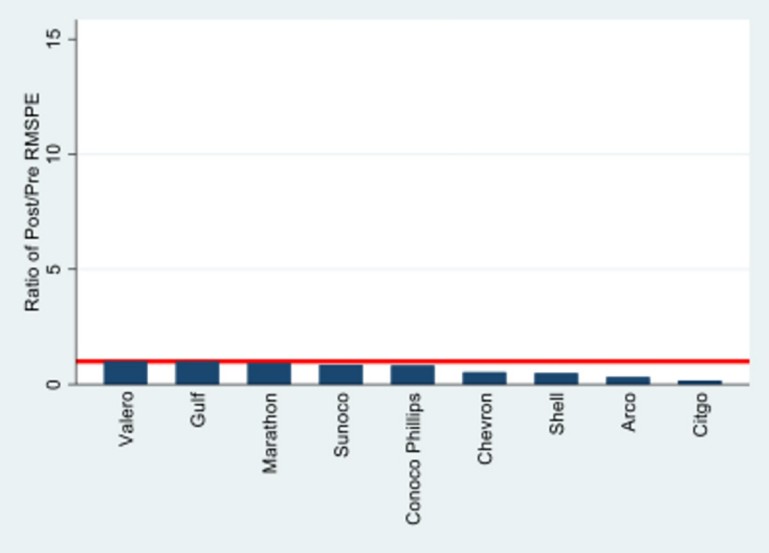

**Fig 4. RMSPE ratios for non-BP oil brands.**

Once again, the horizontal line corresponds to a value of one, indicating no difference in post- vs. pre-Deepwater Horizon disaster RMSPE. If BP's reputational shock spilled over to the rest of the industry, we would expect the ratios for other oil and gas brands to be higher than one. The highest ratio we see is for Valero, with a value of 1.01. The lowest is Citgo, with a value of 0.17. Shell, the largest component of BP's synthetic control, exhibits a ratio of 0.49. This indicates that Shell more closely matched its synthetic control *after* the Deepwater Horizon disaster, a strong indication that there were no spillover effects that biased our results. The low level of RMSPE ratios among all of the oil and gas brands indicates the absence of any significant reputational spillovers within the oil and gas industry.

What might explain the apparent absence of reputational spillovers? Did the industry mount a concerted effort to assure stakeholders through say creating a self-regulatory program? Did individual oil and gas firms take some unilateral action? At the industry level, while the industry did not create a voluntary program, Chevron and other oil and gas companies founded the Marine Well Containment Company, which was tasked to provide a rapid response fleet to contain oil spills. The American Petroleum Institute (API) improved its safety standards and made them available to the public to show the standards in place to promote safety [3, 27, 28]. However, the 2010 annual reports of several oil and gas companies do not show any concrete actions other than a review of their own safety systems and some perfunctory distress about the accident.

## Changes in stock market returns

A firm's reputation is one of its most important intangible assets, which suggests that the negative reputational shock documented in the previous section should have (among other things such as regulatory and legal penalties) serious financial implications for BP. Yet one might argue that citizen perceptions about BP might diverge from those of stock market actors who influence the stock market returns. After all, stock analysts are supposed to have deep expertise about the industry and therefore are in a superior position to assess how an industrial accident might influence future stock returns. But if the Deepwater Horizon disaster led to a consumer boycott or expensive regulatory or legal penalties, then it would affect BP's profitability and

dividends. We should then see BP's stock price decline after the disaster. On the other hand, stock analysts might assess the implications of the Deepwater disaster on future dividends differently. They may not view this disaster as affecting BP's long-term financial health. If so, this disaster would not affect the stock market returns of the company in the long term.

As per the classic Miller-Modigliani model (1961), in competitive financial markets, a firm's stock price represents the net present value of expected future dividends, or the value investors will receive for owning the company. Scholars report mixed evidence on the duration of stock price decline beyond two days. Kaplanski and Levy (2010) find that aviation disasters lead to an average decline in market capitalization of more than $60 billion per disaster [29]. However, after 2 days, the prices reverted to the normal level. In contrast, Capelle-Blancard and Laguna's (2010) study of explosions in chemical plants and refineries finds that stock prices declined by 1.3% over two days and there was a further decline of 12% over six months [30]. Our paper is among the very few to investigate the long-term (7 years) consequences of an industrial accident on stock market returns.

We estimate the effect of the Deepwater Horizon spill on BP's stock market returns using the same synthetic control approach described above. With this, we follow others, like Acemoglu et al. [31], who have applied the synthetic control method to model the counterfactual in the fields of economics and finance [26, 31–37]. The synthetic control method is a novel approach and variation on other event study methods widely used in finance. It addresses some of other event study method's shortcomings [31, 37]. The synthetic control method applies specific weights for each unit based on similarities in variables that are driving the outcome variable of interest, making sure the counterfactual is as similar as possible to the treated unit [31]. The synthetic control method is comparable to a control portfolio. The companies selected for the control portfolio, when using the synthetic control method, are those that are most similar to BP in the underlying characteristics that drive the stock market returns. This data-driven creation of a counterfactual is a more nuanced approach than for example using the market as a counterfactual. An added advantage is that for making causal inferences weaker assumptions are needed than in the traditional event study methods [38].

Once again, the key question is how to identify an appropriate counterfactual for BP–a firm that shows us what BP's stock market returns would have been if the Deepwater Horizon disaster had not happened. We construct a synthetic control for BP using firms that were part of the S&P 500 in 2010, the year of the disaster. Our synthetic control is constructed by matching BP with a weighted average of other S&P 500 firms in terms of their total assets, gross profits, total debt, return on investment, volatility, and "broker recommendation" score that ranges from zero (lowest) to five (highest). All of these variables, along with the stock market returns, were gathered from S&P's Capital IQ database. Unlike our reputational measures, our measures of financial performance are generally not constrained to vary within the same range of -100 to +100. To adjust for differences in initial levels, we normalized our measures of stock market returns, total assets, gross profit, and total debt to be equal to 100 in the first month of our data set, February 2007.

Using these data, we construct a synthetic control for BP based on 187 firms that were listed in the S&P 500 at the time of the Deepwater Horizon disaster. A comprehensive list, including weights (analogous to Table 1) is presented in S1 Appendix. The large number of components makes it difficult to compare BP to any individual firm, but we can compare the mean pre-disaster levels of our predictors between BP and its synthetic control, also known as the "predictor balance" [33]. The predictor balance is shown in Table 2:

These results suggest that our synthetic control is a good match for BP's pre-disaster stock market returns. Since this is a nonparametric estimation technique, we cannot perform a simple hypothesis test to see if there are significant differences between BP and its synthetic

**Table 2. Predictor balance for financial performance.**

|  | BP | Synthetic Control |
|---|---|---|
| Total assets | 108.59 | 108.69 |
| Gross profit | 102.34 | 102.5 |
| Broker Recommendation | 2.12 | 2.12 |
| Total debt | 126.09 | 125.13 |
| Return on investment | 8.32 | 8.35 |
| Volatility | 28.29 | 28.42 |

control after the Deepwater Horizon spill. As with our reputation analysis, we can simply look at the difference in stock market returns between BP and the synthetic control after the disaster. Fig 5 illustrates our results:

The figure above shows that the total returns for BP dropped abruptly below that of our synthetic control after the Deepwater Horizon disaster, which is indicated by the vertical line. This suggests that along with the reputational damage, the disaster also affected BP's financial performance. Although BP was able to return to the initial level of total returns (normalized to 100) by the end of our sample period, its counterfactual had grown by roughly 200% over the same period. However, it is important to perform robustness checks before we draw any conclusions from this comparison, as we do below.

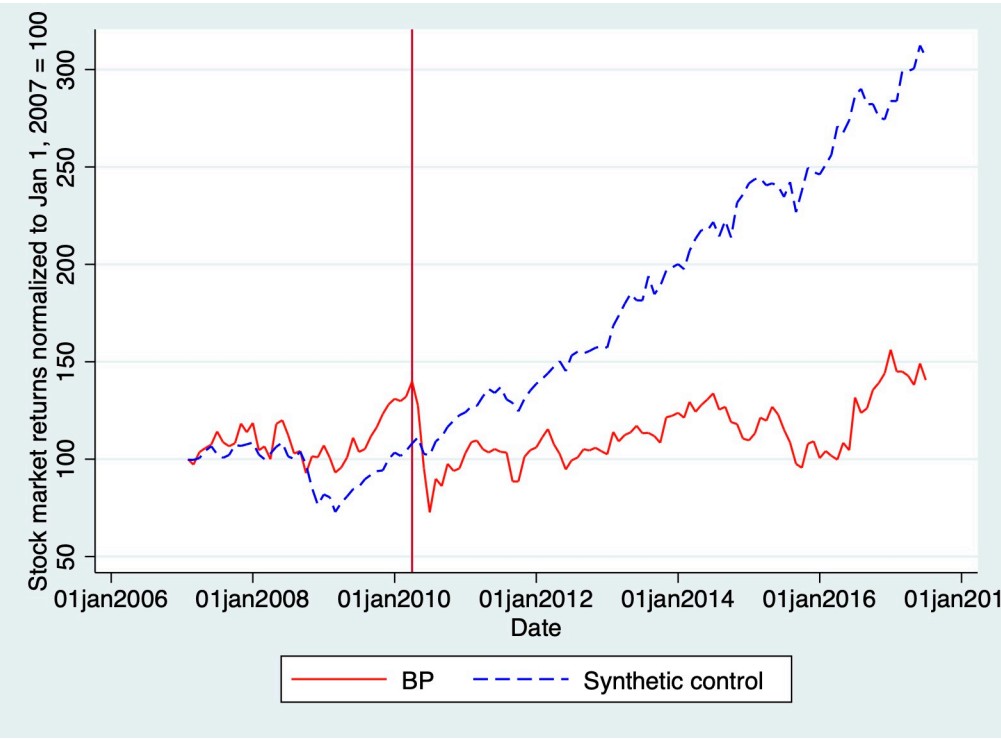

**Fig 5. Total Returns of BP vs. synthetic control.** Note: In S1 Appendix, we have added the figure with the synthetic control without oil and gas companies. There is a necessary tradeoff here; restricting the set of possible components for the synthetic control may help address concerns about contagion but will also reduce the fit of the model. This is why we include oil and gas companies in the original synthetic control and test for spillovers in our robustness checks. Moreover, excluding oil and gas firms from the synthetic control yields essentially the same result as our original model. We also added a figure with the synthetic control based only on oil and gas companies, as comparison.

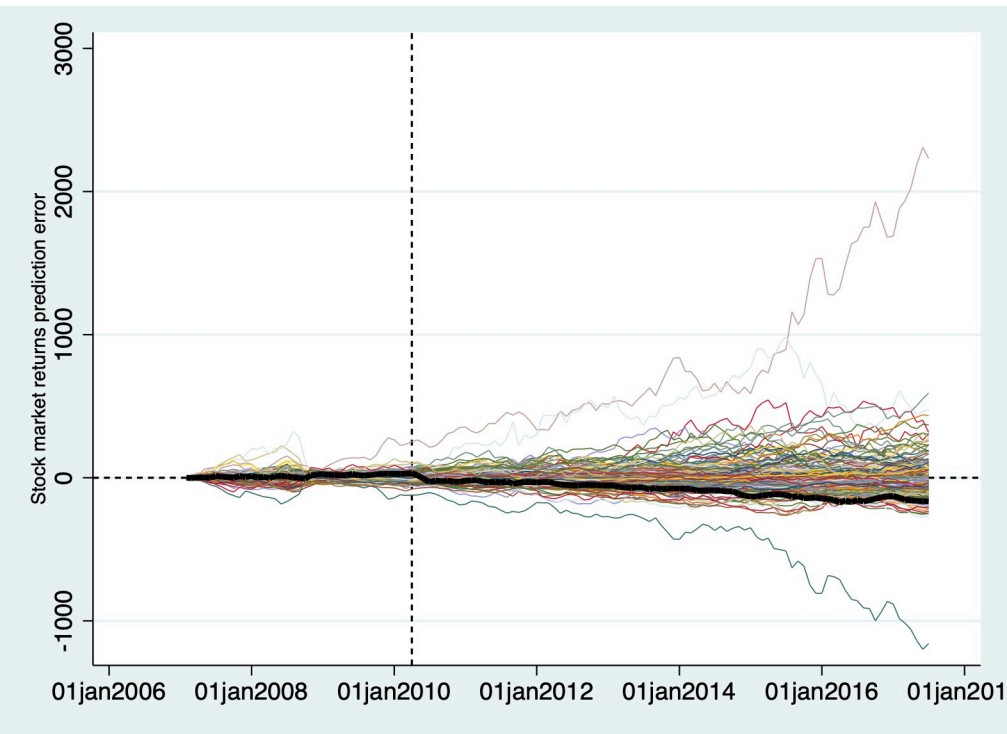

**Fig 6. Total return prediction error for BP vs. synthetic control components.**

### Robustness

The challenge is to be sure that our synthetic control is a good approximation of BP's performance had the Deepwater Horizon disaster never happened. As we did in our reputational analysis, we do this by estimating synthetic control models for each component of BP's synthetic control. Since there are 187 components, we are not able to present a comprehensive set of Post/Pre RMSPE ratios, as we did in Fig 3. In Fig 6 below, we plot the total returns prediction error for BP and all of its synthetic control components, analogous to what was presented in Fig 2.

Ideally, BP (represented by the larger black line) would show a large negative prediction error, and all other components would have prediction errors near zero. Instead, we find a wide variation in prediction errors among our components, some even more negative than BP. If a significant number of the components of BP's synthetic control also experienced large drops in their stock market returns after the Deepwater Horizon disaster, then we cannot be confident that we have identified the effect of the disaster on BP's stock market returns.

Cunningham (2021) suggests computing the Post/Pre RMSPE ratio for the treated unit and each component of the synthetic control and seeing where the treated unit ranks in that distribution [39]. Table 3 shows BP's ranking in the distribution of RMSPE ratios. We calculated this ranking for our full sample period, the two years following the disaster ("Short-term"), and the period two to seven years after the disaster ("Long-term"):

The ranking can also be used to calculate a p-value, which tells us the likelihood that we would observe BP's RMSPE ratio if the disaster had no effect on its total returns. The p-value is simply the rank divided by the total number of firms, which is 187 in BP's case. The results indicate that we cannot reject the null of no effect on stock market returns. We checked the effect both for the two years immediately following the disaster ("Short-term") and for a longer-term period of seven years.

**Table 3. BP post/pre RMSPE ratios and ranks.**

|  | Full Sample (May 2010—July 2017) | Short-Term (May 2010—June 2012) | Long-Term (July 2012 –July 2017) |
|---|---|---|---|
| Post/Pre RMSPE Ratio | 5.84 | 1.65 | 6.79 |
| Ratio Rank (out of 187) | 37 | 90 | 37 |
| P-value (implied by rank) | 0.20 | 0.48 | 0.20 |

We also checked the robustness of our findings by using the more conventional analysis of abnormal returns. We use the capital asset pricing model (CAPM) to predict BP's stock market returns had the Deepwater Horizon oil spill never happened. The capital asset pricing model uses the risk-free rate, BP's beta, which captures the volatility of the stock vis-à-vis the market (in this case the S&P 500) and the expected market return (that of the S&P500). This is captured in the following formula:

$$\bar{R}_a = R_f + \beta(R_m - R_f)$$

For the risk-free rate we use the US 10-year treasury bond, as is customary, and for the beta, we used BP's two-year market beta. These data were obtained from the CapitalIQ database.

Subsequently, we compare the expected returns with BPs actual returns to calculate the abnormal returns. The abnormal returns are the actual returns, minus the returns we predicted using CAPM. Fig 7 shows the predicted and the actual returns. Showing that, although there is an immediate drop in market returns after the disaster, there is no clear break in the trend from before the disaster.

Fig 8 plots the abnormal returns of BP. Which is the difference between the actual market returns and the expected market returns based on CAPM.

More formally, we use a simple regression model to see whether the abnormal returns are statistically significantly different after the disaster compared to before the disaster. Estimation results are reported in Table 4. We check for short-term (2 years) and long-term (2–7 years) differences, separately. We find no significant differences. We run the same analysis also excluding other integrated oil and gas companies and only oil and gas companies, finding similar results, which are reported in S1 Appendix. This means that we fail to reject the null that BP experienced no change in abnormal returns after the disaster. Though the decline in stock returns in the wake of the disaster is not statistically significant, we recognize that some might consider it to be economically significant (about 27% decline over a two-year period).

We also compare the cumulative abnormal returns (CAR) of BP with those of other integrated oil and gas companies. We find initial negative abnormal returns for BP in the first month after the disaster, but these stabilize over time and then slowly diminish in the long run. This, however, is in line with the cumulative abnormal returns for other integrated oil and gas companies as shown in Fig 9. The initial drop however is not recovered over time, and still signals an economically important drop in BP's value.

**Spillover effects.** Did the Deepwater Horizon oil spill influence the total returns of other firms in the oil and gas industry? Fig 10 gives a general impression of the behavior of the market returns of other oil and gas companies in the aftermath of the disaster. It also shows how BP's performance may have dropped vis-à-vis other oil and gas firms, despite the effect of the oil spill not being statistically significant.

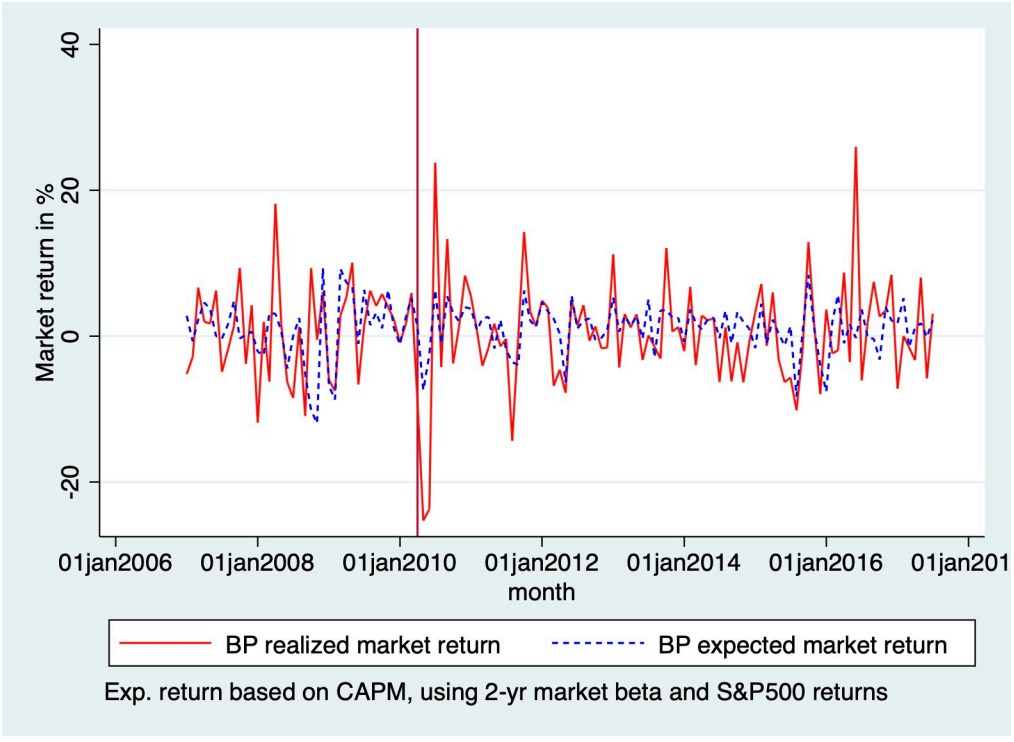

**Fig 7. CAPM expected market returns vs. actual market returns BP.**

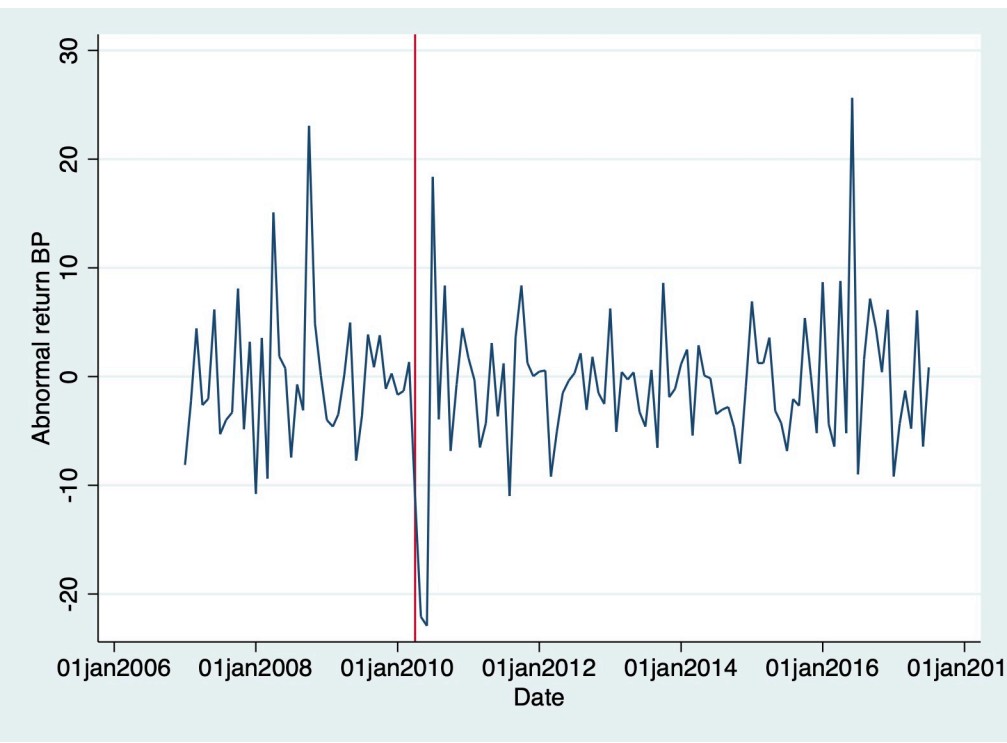

**Fig 8. BP's abnormal returns.**

**Table 4. BP abnormal returns pre vs post disaster.**

| | (1) |
| --- | --- |
| | Abnormal Returns |
| Short term (0–2 years) | -1.132 |
| | (0.577) |
| Long term (2–7 years) | -0.569 |
| | (0.722) |
| Constant | -0.179 |
| | (0.887) |
| Observations | 125 |

p-values in parentheses: * p<0.05, ** p<0.01, *** p<0.001

We also estimated synthetic control models for all of the S&P 500 firms in the Oil & Gas Exploration & Production and Integrated Oil & Gas GICS sub-industries. Although Shell is not included in the S&P 500, we included it in our analysis because it was one of the brands included in our reputation analysis. We report the RMSPE ratios and associated p-values in Table 5.

Spillover effects would be indicated by RMSPE ratios greater than one in Table 5. According to the results in Table 5, other Oil & Gas firms exhibited RMSPE ratios greater than one during our sample period, but we must also consider the significance of these results. As we did with BP, we calculated the rank of each firm's RMSPE ratios among the components of their respective synthetic controls. The values in parentheses in Table 5 are the p-values calculated based on those rankings. We find no evidence of significant spillover effects on the stock

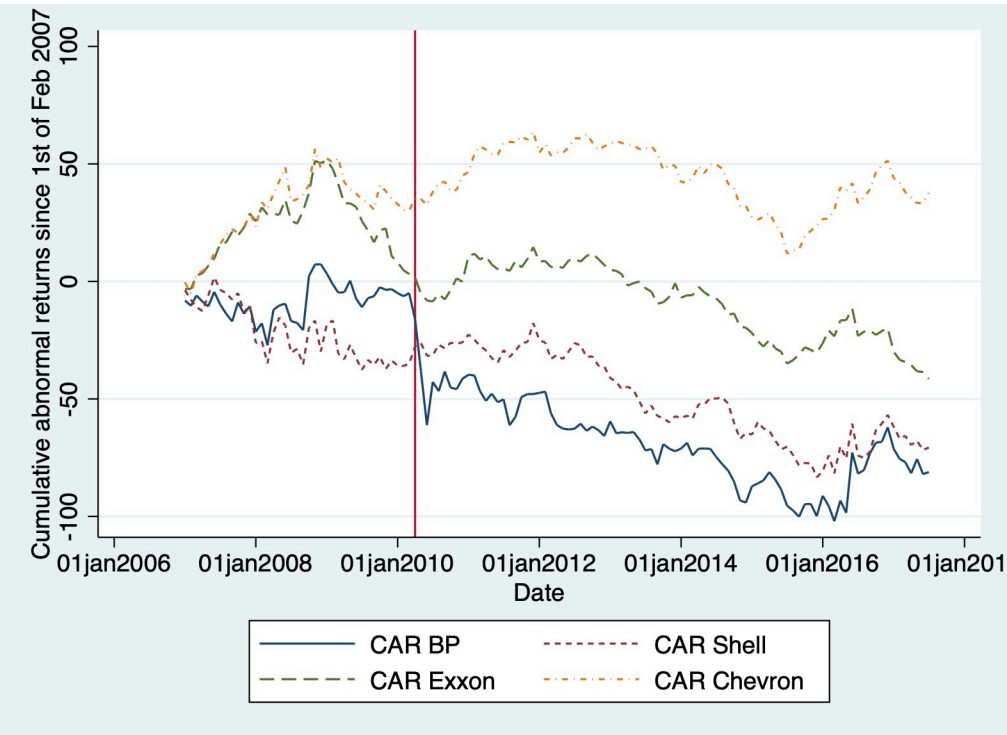

**Fig 9. Cumulative Abnormal Returns (CAR) for BP, Shell, Chevron, and ExxonMobil.**

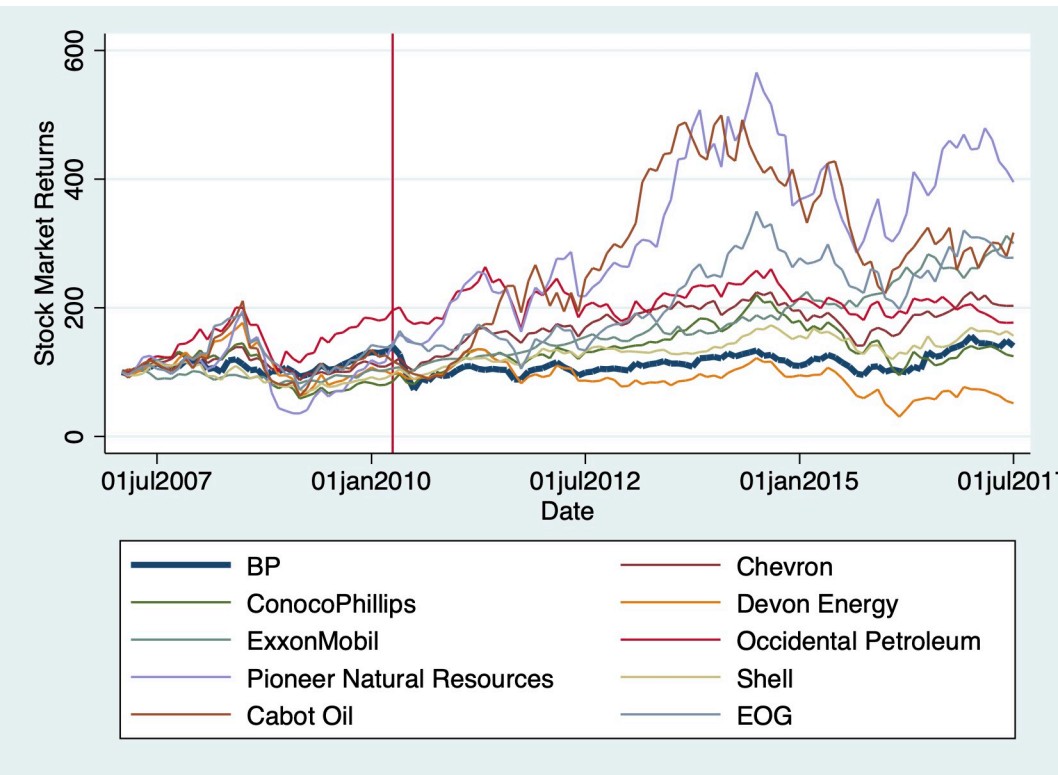

**Fig 10. Market returns of oil and gas firms after deepwater horizon oil spill.**

market returns of other Oil & Gas firms. We interpret these results to mean that the Deepwater Horizon disaster did not spill over to stock market returns of other firms in the same industry.

We should note that this null result may reflect the steps other oil and gas firms took toward mitigating disasters after Deepwater Horizon. This response might have been motivated by the desire to preempt new regulations which could impact the sectors' financial market performance, as Baron and Diermeier have argued [40]. If these steps were taken in anticipation of future effects on financial performance, they could bias the estimates presented here. However, we do not find much evidence of the evolution of any industry-level voluntary program, which would lead to an ex-ante response to mitigate accidents from happening in the first place. Recall the chemical industry launched its self-regulatory program, Responsible Care, in the aftermath of the 1984 Union Carbide's Bhopal disaster. It outlined best practices that it expected its member firms to follow to prevent chemical disasters from taking place in the first place. Yet, no such ex-ante industry-level response emerged in the aftermath of Deepwater.

Also, Shell, while professing deep distress, defended offshore drilling as necessary to meet global demand and suggested that if it had been in charge, the accident would not have happened. Exxon Mobil and Chevron also followed the same playbook and took no substantial unilateral action.

What might explain the lack of long-term consequences for BP's financial performance, given the plethora of risks resulting from the accident? BP's 2012 annual report acknowledged:

The Gulf of Mexico oil spill has damaged BP's reputation, which may have a long-term impact on the group's ability to access new opportunities, both in the US and elsewhere. Adverse public, political and industry sentiment towards BP, and towards oil and gas

**Table 5. Pre/post RMSPE ratios and P-values for oil & gas firms.**

| Name | Full Sample (May 2010—July 2017) | Short-Term (May 2010—June 2012) | Long-Term (July 2012 – July 2017) |
|---|---|---|---|
| BP | 5.84 | 1.65 | 6.79 |
| | (0.20) | (0.48) | (0.20) |
| Cabot Oil & Gas | 5.46 | 2.41 | 6.24 |
| | (0.21) | (0.26) | (0.21) |
| Chevron | 2.78 | 1.43 | 3.15 |
| | (0.59) | (0.56) | (0.58) |
| ConocoPhillips | 2.06 | 0.48 | 2.41 |
| | (0.72) | (0.94) | (0.72) |
| Devon Energy | 2.65 | 0.37 | 3.10 |
| | (0.57) | (1.00) | (0.57) |
| EOG Resources | 1.63 | 1.13 | 1.79 |
| | (0.82) | (0.70) | (0.82) |
| ExxonMobile | 6.18 | 0.99 | 7.24 |
| | (0.31) | (0.77) | (0.31) |
| Occidental Petroleum | 1.17 | 1.63 | 0.94 |
| | (0.92) | (0.48) | (0.96) |
| Pioneer Natural Resources | 2.57 | 1.67 | 2.84 |
| | (0.61) | (0.48) | (0.62) |
| Shell | 6.69 | 0.87 | 7.84 |
| | (0.15) | (0.81) | (0.14) |

Note: P-values in parentheses are calculated based on the firm's rank among the components of its own synthetic control estimator. *** $p<0.01$, ** $p<0.05$, * $p<0.1$

Arco, Citgo, and Gulf are not represented in Table 5 because no stock market returns data were available. Arco has been a subsidiary of BP since 2000. Citgo is privately held, and Gulf merged with Chevron in 1985. Stock market returns data were available for Sunoco, but they exited the oil refining business in 2010, and so may not be an appropriate comparator for BP.

drilling activities generally, could damage or impair our existing commercial relationships with counterparties, partners and host governments and could impair our access to new investment opportunities, exploration properties, operatorships or other essential commercial arrangements with potential partners and host governments, particularly in the US" [41].

The financial penalties BP incurred were substantial costs: $60 billion (in relation to BP's market capitalization of about $170 billion in 2009) as fines, penalties, clean up and mitigating costs. Arguably, because BP took several steps to mitigate the reputational damage and change internal systems, these might have provided some cushion against a decline in stock market returns. BP's response took place at multiple levels. In terms of public relations, it issued a series of public apologies. Its so-called "apology commercial", featuring its CEO Tony Hayward, was criticized because it tended to highlight what BP had done in the past, instead of sufficiently and honestly taking responsibility for the oil spill. His Congressional testimony was more contrite, but Hayward seemed not to provide clear answers in response to questions posed by the Congressional committee. All in all, it is remarkable that BP suffered no significant decrease in stock market returns.

## Conclusions

Our results have important implications for the literature on corporate environmental governance. We demonstrate that, beyond the obvious legal liability, citizens will hold firms accountable for their workplace safety and environmental records for a substantial period. Recall that since 2000, BP had invested a substantial sum in the "Beyond Petroleum" campaign to highlight its commitment to environmental protection. Further, after the Deepwater accident, it invested about $500 million in brand enhancement [42]. Yet, the reputational damage caused by the Deepwater accident has persisted. It needs to be mentioned however that the impact of a firm's workplace safety and environmental conduct on its reputation is mediated by the attention it gets from media and activists. The scale of the Deepwater Horizon oil spill was so large that the issue got automatic visibility. For smaller accidents, reputational damages could increase if environmental groups and the media decide to focus on it in their campaigns.

Nevertheless, this finding should be a wake-up call for any firm as it develops its workplace safety and environmental management strategies. Information about firms' safety records seems to have a lasting effect on their reputation. In addition, this research provides insights on reputational spillovers among firms selling an undifferentiated product and can help us understand under which conditions firms within the same sector hold their reputation in common.

While reputational effects are long-lasting, the financial markets appear to bounce back rather quickly after an initial shock. Brand reputation and financial market returns are driven by different mechanisms. Reputations could be influenced by consumers' product experience and value perceptions, while stock market returns could be affected by profits, new product launches, and market volatility. Arguably brand reputations could certainly affect stock market returns but the opposite effect is less plausible. Factors such as industrial accidents or regulatory scrutiny could affect both. In this case, BP seems to have escaped any long-term consequences of its environmental disaster in terms of its stock market returns. This is in some ways disappointing: a major disaster should severely penalize the company's financial market returns for a long time. Because corporate compensation is often linked with the stock price, top management will pay serious attention to the issue of industrial safety if their compensation is affected. Of course, in the aftermath of Deepwater, BP's CEO resigned, and BP paid several billion dollars in penalties. Yet, we did not find a statistically significant effect on its stock market returns, which calls into question whether stock markets create sufficiently strong incentives for firms to pay careful attention to industrial safety.

The lack of a coordinated industry response to the disaster is also puzzling. Why might this be so? There are several possible reasons. First, this accident did not result in large-scale death or dislocation of communities (as in wildfires or hurricanes). Thus, after the initial shock and the graphic images of oil pollution and the destruction of marine life, the media tended to move on. Indeed, within 6 months of the accident, Louisiana (the state most impacted by the spill) politicians were demanding that the federal government should not over-regulate this industry. Mary Landrieu, the Democratic senator from Louisiana, demanded that the EPA lifts the ban on BP from securing federal contracts [43]. Interestingly, even the UK government lobbied the Obama administration that BP should not be forced to pay excessive compensation.

Moreover, environmental issues have become deeply partisan. As environmental groups saw the Deepwater accident as an opportunity to push the climate agenda by demanding an end to offshore drilling, partisan identities flared up. Conservatives rushed to defend the offshore oil industry—and during Trump's Presidency, even some of the modest new regulations were rolled back. Thus, bipartisan efforts to hold the oil industry accountable were weak. A

new federal regulation (based on the recommendations of the US Chemical Safety Board), the 2016 Wells Control Rule, emerged. The federal government created a new regulatory body, the Bureau of Safety and Environmental Enforcement (located in the Department of Interior), which is tasked with frequently inspecting offshore facilities for regulatory compliance. However, as Republicans took over the House in 2011, it was clear that new stringent federal regulations aimed at this industry will be difficult to enact. This might also be why the industry did not feel the need to invest in regulatory preemption via voluntary programs. This lack of industry-level efforts has not been systematically explored and requires future research.

Lastly, the lack of industry response could exactly be due to the lack of financial consequences of disasters. Several industry associations have sponsored voluntary programs that outline best practices, codes of conduct and so on. Scholars suggest that these industry-level clubs [23, 24] are vehicles to provide collective insurance, particularly in the context of industrial accidents; the assumption being that a mishap for one firm creates negative reputational spillovers across the industry. The classic case is the chemical industry's Responsible Care Program whose emergence was partially motivated by the 1984 Bhopal disaster at Union Carbide's facility [44]. Thus, via industry-level clubs, industry associations can build collective goodwill for environmental stewardship and secure some sort of reputational insurance for their members. Yet, our paper shows that even a major industrial accident such as Deepwater Horizon may not cause significant declines in stock market returns for oil and gas firms, despite long-term reputational damage. Hence, our finding challenges the fundamental motivation for industry associations to create industry-level regulatory clubs.

## Supporting information

**S1 Appendix.**
(DOCX)

**S1 Dataset.**
(DOCX)

## Acknowledgments

Earlier versions of this paper were presented at the Environmental and Politics Governance Conference 2020 and Western Political Science Association Conference 2021. We'd like to thank the reviewers and the panel for their feedback.

## Author Contributions

**Conceptualization:** William McGuire, Ellen Alexandra Holtmaat, Aseem Prakash.

**Data curation:** William McGuire, Ellen Alexandra Holtmaat, Aseem Prakash.

**Formal analysis:** William McGuire, Ellen Alexandra Holtmaat, Aseem Prakash.

**Funding acquisition:** William McGuire, Ellen Alexandra Holtmaat, Aseem Prakash.

**Investigation:** William McGuire, Ellen Alexandra Holtmaat, Aseem Prakash.

**Methodology:** William McGuire, Ellen Alexandra Holtmaat, Aseem Prakash.

**Project administration:** William McGuire, Ellen Alexandra Holtmaat, Aseem Prakash.

**Resources:** William McGuire, Ellen Alexandra Holtmaat, Aseem Prakash.

**Software:** William McGuire, Ellen Alexandra Holtmaat, Aseem Prakash.

**Supervision:** William McGuire, Ellen Alexandra Holtmaat, Aseem Prakash.

**Validation:** William McGuire, Ellen Alexandra Holtmaat, Aseem Prakash.

**Visualization:** William McGuire, Ellen Alexandra Holtmaat, Aseem Prakash.

**Writing – original draft:** William McGuire, Ellen Alexandra Holtmaat, Aseem Prakash.

**Writing – review & editing:** William McGuire, Ellen Alexandra Holtmaat, Aseem Prakash.

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
