## [Decision Letter · Decision Letter 0]

21 Jul 2021

PONE-D-21-16006

Penalties for industrial accidents: The impact of the Deepwater Horizon accident on BP's reputation and stock price

PLOS ONE

Dear Dr. Holtmaat,

Thank you for submitting your manuscript to PLOS ONE. After careful consideration, I feel that it has merit but does not fully meet PLOS ONE’s publication criteria as it currently stands. 

I consider that this is an interesting manuscript for Plos One and I find clear and well-written. Reviewers have provided a wide number of constructive comments. Most of them are minor remarks but one of the reviewers have shown major concerns regarding to the methodological aspects of the manuscript.

Summin up, I think that the manuscript will be enriched if the authors are able to address all reviewers’ comments, but I am particularly worried because of the doubts of one of the reviewers with the methodology used as well as the limitations of the data source.

Therefore, we invite you to submit a revised version of the manuscript that addresses the points raised during the review process.

We look forward to receiving your revised manuscript.

Kind regards,

J E. Trinidad Segovia

Academic Editor

PLOS ONE

Journal Requirements:

Reviewers' comments:

Reviewer's Responses to Questions

**Comments to the Author**

1. Is the manuscript technically sound, and do the data support the conclusions?

Reviewer #1: Yes

Reviewer #2: No

Reviewer #3: Yes

2. Has the statistical analysis been performed appropriately and rigorously? 

Reviewer #1: Yes

Reviewer #2: No

Reviewer #3: Yes

3. Have the authors made all data underlying the findings in their manuscript fully available?

Reviewer #1: No

Reviewer #2: No

Reviewer #3: Yes

4. Is the manuscript presented in an intelligible fashion and written in standard English?

Reviewer #1: Yes

Reviewer #2: Yes

Reviewer #3: Yes

5. Review Comments to the Author

Reviewer #1: I love your paper and your findings completely make sense to me. Three quick questions.

1. Why didn’t you apply the event study method for stock price analysis?

2. I am in sync with you that some scandal did not necessarily sully the reputation of other similar companies (e.g. the Volkswagen Diesel scandal did not affect other German car companies). One quick question that pops up in mind is the BP scandal could have the reverse effect on its competitors: the competitors’ stock prices might have risen in the wake of the oil spill. Could you check one more time whether the stock prices of BP’s competitors rose after the DeepWater Horizon? I am eager to see the graph of each competitor’s stock prices (APA Corp, Cabot, Chevron, Conoco Phillips, Devon Energy, EOG resources, Exxon Mobile, Marathon, Shell, and Valero) before/after the Deepwater Horizon. Could you display a list of graphs showing historical stock prices of each competitor?

3. I am really curious about the correlation between BP’s reputation and its stock prices. Could you please check the correlation between the two variables in separate time windows (before/after the Deepwater Horizon). Do they have sig relationships or not? It would be very interesting if we could see a more dynamic relation between the two variables (e.g., BP’s reputation might have had positive relationships with its stock prices for 1-2 months after the Deepwater Horizon scandal (i.e., both of them dropped after the negative event), but not 6 months later (i.e., BP’s reputation still suffered, but not its stock prices).

Reviewer #2: Penalties for industrial accidents: The impact of the Deepwater Horizon accident

on BP’s reputation and stock price

6/29/2021

Main point of this paper

Paper attempts to measure long term impact to BP after the 2010 Deepwater Horizon explosion using two metrics: the reputation variables obtained from YouGov’s Brand Index and financial data from Capital IQ. They develop a proxy for BP’s reputation and find a 50% decline after the accident and a persistent decline until 2017. When testing effect on stock prices, they only find a short-term effect. They conclude that even though reputation suffers in the long run, stock prices do not suffer in the long run.

Methodology issues

Some clarifications are needed to fully understand the data and methodologies used.

Some suggestions are provided for the comparison of stock performance.

Reputation

Data source and its limitations must be clearly stated.

YouGov data must be described and brands in the energy sector must be listed. The use of brands instead of companies must be clearly explained and when applicable, connect the brand with the firm, and state when brands belong to non-public firms.

Industries within the energy sector then must be clearly separated showing the main differences among them. BP is an integrated Oil and Gas company, like Chevron, Exxon and Shell. Firms like Marathon and ConocoPhillips are considered Exploration and Production while Valero, Sunoco, Citgo are in the refining and marketing of gasoline. An extremely important omission in this study is the absence of oil and gas services companies like Schlumberger, Halliburton, Transocean, Weatherford and others, which were directly involved in the accident. The absence, though caused by the data source, must be recognized as one of the weakness of the study.

More detail is needed on how the weights for the “control” brand are determined. For example, you start with a full sample of firms or a subsample? What’s the sample size? To be a candidate for inclusion in the synthetic control, do you remove oil & gas firms where suspected spillover occurs? How many firms are left? What is the technique used? Does the 0.756 for Shell means that the synthetic control is 75% Shell? Why do Marathon and Chevron not appear in the synthetic even though they are in the same industry?

Stock price

Methodology is not consistent with financial literature. At first glance it appears that they just want to adapt the methodology used when evaluating reputation to stock prices.

Evaluation of the performance of a company’s stock is normally done in total annual returns, which include price appreciation and dividend income.

If you concentrate on the performance of one firm, it must be compared against a portfolio of firms with similar characteristics. So, the formation of a good comparable portfolio is important. However, such portfolios must be constructed using meaningful financial ratios/variables, such as total assets, profitability, leverage, efficiency and risk. Authors used 6 ratios to match firms however only ROA and Broker recommendation can be reasonably used to match firms. Per share ratios are not commonly used.

Authors say 273 firms are used to build a synthetic control. Those are just too many firms for a “match”, especially if they want to match the risk of the company in question. if they want to use a broad index, they can just use the S&P 500 as a measure of the market and compare the firm against the overall US market.

Using data publicly available I make a quick comparison of the performance from May 2010 to July 2017 of BP vs the S&P500 index. Average annual return for BP is around 2.4% for the 7 years after the accident, while the market had an average annual return around 23%. That is an economically significant difference, and the numbers suggest that BP has not caught up with the normal behavior of the stock market, even after considering BP’s market beta.

Risk was not considered in this analysis. The market model is normally used in the literature and market-adjusted or beta-adjusted returns can be obtained when making a risk-adjusted comparison.

In summary. share price prediction is not a common practice in the literature and the RMSPE comparison is not convincing. It is suggested that authors create portfolios and then make point comparisons of CAR (cumulative abnormal return) and/or time series of monthly total returns (not prices) and then make comparisons against meaningful portfolios. Possible portfolios to use for comparison would be based on oil & gas industries (integrated, services, refining, exploration). To test the spillover effect, build portfolios with companies not in the energy sector but with equal risk on the period before 2010.

Reviewer #3: You are looking at an interesting topic, both for the management of environmental issues by firms, but also, more generally, for private regulation and the reduction of environmental damages.

In that spirit, the reputational implications are indeed fundamental, and the originality of your work is to focus on one specific case but also to use quantitative methods to do so. Overall, I like what you do in the paper and I find your analysis both interesting and convincing. My comments are thus mostly developmental.

One caveat, both in your empirics and in your interpretation of the results has to do with what firms do when they see a large damage happening. Contrary to what you suggest in your paper, both rival firms and industry associations should have incentives to self-regulate as much as possible in order to prevent both other spillages and reputation losses. This might be one of the key reasons why you don’t observe much regarding spillovers to other firms or long-term effects on stock-prices. Econometrically, this raises an important question as there is a set of key unobserved variables that have to do with what firms do when they observe BP’s pollution problem. Can you do something about this? Since you don’t have many firms to cover, it might be possible to collect some data on what industry rivals have reacted. It would strengthen your analysis for sure.

The biggest issue, though, is in the interpretation of the results. You argue in the Discussion section that the industry association plays no role but it might be the opposite: all firms might have reacted fast and in a coordinated way, and might thus have pre-empted the problem. This might be one of the reasons why there is no spillover to the rest of the sector. In that case, the industry association would both play a role in pushing towards self-regulation and in protecting the firms.

Note that this pre-emption argument is pretty much what the theoretical literature would predict. See for instance Baron and Diermeier in the Journal of Economics and Management Strategy.

Another question is that there are key players that are currently not accounted for in your paper: environmental activists and the media. These typically play an important role in creating saliency regarding the issue, thus impacting firms’ reputations. On this, see for instance the article by Breitinger and Bonardi on reputational damages in Business and Politics. There might be heterogeneity regarding environmental issues that way. Of course, one could argue that for a significant issue such as the one you are looking at here, this might be less relevant (since the issue is going to be very salient in any case). But I think you should at least mention this.

Last point that, I think, should be discussed is the difference between reputation and stock prices. At the moment, you treat these as the same thing and you hint that their evolution is driven by similar mechanisms. This would be an overstatement. I think you need to take these differences into account in the discussion of your results.

6. PLOS authors have the option to publish the peer review history of their article (what does this mean?). If published, this will include your full peer review and any attached files.

Reviewer #1: No

Reviewer #2: No

Reviewer #3: No

---

## [Author Response · Author response to Decision Letter 0]

25 Jan 2022

Rebuttal Memo

Penalties for industrial accidents: The impact of the Deepwater Horizon accident on BP’s reputation and stock market returns 

(previous title: Penalties for industrial accidents: The impact of the Deepwater Horizon accident on BP’s reputation and stock price) 

(PONE-D-21-16006).

Dear Reviewers:

Thank you for your thoughtful, detailed, and constructive comments on our manuscript. We agree with the substance of your concerns and are encouraged by the strengths you see in the manuscript and the contributions our paper can make. We enclose a memo detailing your suggestions (in italics) and outlining how we address them in the revised manuscript 

Regards,

Authors

Reviewer 1

I love your paper and your findings completely make sense to me. Three quick questions.

1. Why didn’t you apply the event study method for stock price analysis?

Response: 

We include the event study method as a robustness check in the revised paper. Our substantive results do not change.

2. I am in sync with you that some scandal did not necessarily sully the 

reputation of other similar companies (e.g. the Volkswagen Diesel scandal did not affect other German car companies). One quick question that pops up in mind is the BP scandal could have the reverse effect on its competitors: the competitors’ stock prices might have risen in the wake of the oil spill. Could you check one more time whether the stock prices of BP’s competitors rose after the DeepWater Horizon? I am eager to see the graph of each competitor’s stock prices (APA Corp, Cabot, Chevron, Conoco Phillips, Devon Energy, EOG resources, Exxon Mobile, Marathon, Shell, and Valero) before/after the Deepwater Horizon. Could you display a list of graphs showing historical stock prices of each competitor?

Response

Great point. We have included a new graph in the revised paper. As the graph shows, there is no consistent trend in competitor firms’ total or stock market returns (that Reviewer 2 wants us to focus on) in the aftermath of the accident.

3. I am really curious about the correlation between BP’s reputation and 

its stock prices. Could you please check the correlation between the two variables in separate time windows (before/after the Deepwater Horizon). Do they have sig relationships or not? It would be very interesting if we could see a more dynamic relation between the two variables (e.g., BP’s reputation might have had positive relationships with its stock prices for 1-2 months after the Deepwater Horizon scandal (i.e., both of them dropped after the negative event), but not 6 months later (i.e., BP’s reputation still suffered, but not its stock prices).

Response

Excellent suggestion. Please see the graphs below.

BP’s reputation and total returns (as well as stock prices) appear to move in tandem before and after the accident. 

Reviewer #2

Paper attempts to measure long term impact to BP after the 2010 Deepwater Horizon explosion using two metrics: the reputation variables obtained from YouGov’s Brand Index and financial data from Capital IQ. They develop a proxy for BP’s reputation and find a 50% decline after the accident and a persistent decline until 2017. When testing effect on stock prices, they only find a short-term effect. They conclude that even though reputation suffers in the long run, stock prices do not suffer in the long run.

1. YouGov data must be described and brands in the energy sector must 

be listed. The use of brands instead of companies must be clearly explained and when applicable, connect the brand with the firm, and state when brands belong to non-public firms.

Response

We have included this information in the revised paper. We now have an appendix with all the brands in YouGov data and highlight those brands in the energy sector. 

2. Industries within the energy sector then must be clearly separated 

showing the main differences among them. BP is an integrated Oil and Gas company, like Chevron, Exxon and Shell. Firms like Marathon and ConocoPhillips are considered Exploration and Production while Valero, Sunoco, Citgo are in the refining and marketing of gasoline. An extremely important omission in this study is the absence of oil and gas services companies like Schlumberger, Halliburton, Transocean, Weatherford and others, which were directly involved in the accident. The absence, though caused by the data source, must be recognized as one of the weakness of the study.

Response

We have clarified which GICS sub-industries are included in our analysis, and acknowledged the absence of other relevant firms such as Schlumberger, Halliburton, Transocean, and Weatherford.

3. More detail is needed on how the weights for the “control” brand are 

determined. For example, you start with a full sample of firms or a subsample? What’s the sample size? To be a candidate for inclusion in the synthetic control, do you remove oil & gas firms where suspected spillover occurs? How many firms are left? What is the technique used? Does the 0.756 for Shell means that the synthetic control is 75% Shell? Why do Marathon and Chevron not appear in the synthetic even though they are in the same industry?

Response

We have clarified these issues in the manuscript. 

Sample size: 

For the reputation analysis, the initial sample size was 660 brands, while for stock price analysis, it was 189 companies. 

Subsample: 

Within the full sample, we identify the combination of firms and weights for each firm that would create a synthetic firm with underlying characteristics that are as close as possible to the underlying characteristics of BP before the disaster. For reputation analysis, the underlying characteristics are the general impression of the brand, the perceived quality of the product, the value for money, and the respondent’s willingness to work for the company. For financial analysis, these are total assets, gross profits, total debt, return on investment, volatility, and “broker recommendation” score. 

Excluding oil and gas firms: 

To create synthetic control for BP, we included oil and gas firms in the sample (and then allowed the estimator to identify which specific firms to include in the synthetic control). 

How many firms are left? 

For reputation analysis, the synthetic control consists of 15 (of the 660 in the initial sample) firms, while for the financial analysis, it consists of 188 (of the 189 in the initial sample) firms.

Technique used: 

To identify firms for including in the synthetic control and assigning them weights (to minimize the difference between the underlying characteristics of the synthetic control compared to BP), we use the estimator designed by Abadie et al. Here is the citation: Abadie A, Diamond A, Hainmueller J. 2015. Comparative politics and the synthetic control method. American Journal of Political Science, 59(2):495–510. 

Is synthetic control 75% Shell? 

Yes, it is.

Why are Marathon and Chevron not part of the synthetic control: 

The reason is that the estimator was able to create a closer match with BP’s underlying characteristics by excluding these firms. 

Spillovers: 

In creating the synthetic control for BP, we did not exclude oil and gas firms. In this analysis of the spillover effect, we formed synthetic controls that exclude other oil and gas firms. 

4. Stock Price: Methodology is not consistent with financial literature. At 

first glance it appears that they just want to adapt the methodology used when evaluating reputation to stock prices. Evaluation of the performance of a company’s stock is normally done in total annual returns, which include price appreciation and dividend income.. If you concentrate on the performance of one firm, it must be compared against a portfolio of firms with similar characteristics. So, the formation of a good comparable portfolio is important. However, such portfolios must be constructed using meaningful financial ratios/variables, such as total assets, profitability, leverage, efficiency and risk. Authors used 6 ratios to match firms however only ROA and Broker recommendation can be reasonably used to match firms. Per share ratios are not commonly used.

Response

We have revised our methodology (and the paper title) for the financial analysis. Importantly, we now use total returns (or stock market returns) as our dependent variable and conduct a robustness check by calculating BP’s abnormal returns within the CAPM framework. To construct the synthetic control for financial analysis, we now use total assets, gross profits, total debt, return on investment, volatility, and broker recommendation, as suggested. 

5. Authors say 273 firms are used to build a synthetic control. Those are 

just too many firms for a “match”, especially if they want to match the risk of the company in question. 

Response

We have clarified this issue in the paper. We now start with a sample of 189 firms. This number differs from our previous draft due to missing observations for some variables we now use to construct the synthetic control. To reiterate, not all 189 firms necessarily become part of the synthetic control. The reason is that the estimator creates the synthetic control by selecting specific firms and assigning them weights to minimize the difference between the synthetic control and BP in terms of their (pre-intervention) characteristics. Based on your very helpful suggestions, we have now updated the variables impacting market returns and included risk (volatility) as a variable informing the formation of our synthetic control. 

6. If they want to use a broad index, they can just use the S&P 500 as a 

measure of the market and compare the firm against the overall US market. Using data publicly available I make a quick comparison of the performance from May 2010 to July 2017 of BP vs the S&P500 index. Average annual return for BP is around 2.4% for the 7 years after the accident, while the market had an average annual return around 23%. That is an economically significant difference, and the numbers suggest that BP has not caught up with the normal behavior of the stock market, even after considering BP’s market beta.

Risk was not considered in this analysis. The market model is normally used in the literature and market-adjusted or beta-adjusted returns can be obtained when making a risk-adjusted comparison.

Response

In the revised paper, we have included a robustness check to calculate the abnormal returns and compare BP’s returns to the market returns of S&P 500 companies. We do not find that the abnormal returns are statistically significantly different between pre-disaster and post-disaster time periods. 

7. In summary. share price prediction is not a common practice in the literature and the RMSPE comparison is not convincing. It is suggested that authors create portfolios and then make point comparisons of CAR (cumulative abnormal return) and/or time series of monthly total returns (not prices) and then make comparisons against meaningful portfolios. Possible portfolios to use for comparison would be based on oil & gas industries (integrated, services, refining, exploration). To test the spillover effect, build portfolios with companies not in the energy sector but with equal risk on the period before 2010.

Response:

In the revised paper, we compare BP’s cumulative abnormal returns to other integrated oil and gas companies. We find that the cumulative abnormal returns of BP do go down over time, but more so starting from two years after the oil spill. The same trend is visible, however, in other integrated oil and gas companies, which suggests likely other factors at play in this divergence from the market, particularly given the absence of a short-term negative impact. We have noted this in the revised manuscript.

Reviewer 3

1. You are looking at an interesting topic, both for the management of environmental issues by firms, but also, more generally, for private regulation and the reduction of environmental damages.

In that spirit, the reputational implications are indeed fundamental, and the originality of your work is to focus on one specific case but also to use quantitative methods to do so. Overall, I like what you do in the paper and I find your analysis both interesting and convincing. My comments are thus mostly developmental.

Response

Thank you

2. One caveat, both in your empirics and in your interpretation of the results has to do with what firms do when they see a large damage happening. Contrary to what you suggest in your paper, both rival firms and industry associations should have incentives to self-regulate as much as possible in order to prevent both other spillages and reputation losses. This might be one of the key reasons why you don’t observe much regarding spillovers to other firms or long-term effects on stock-prices. Econometrically, this raises an important question as there is a set of key unobserved variables that have to do with what firms do when they observe BP’s pollution problem. Can you do something about this? Since you don’t have many firms to cover, it might be possible to collect some data on what industry rivals have reacted. It would strengthen your analysis for sure.

Response

This is an excellent point. We mention this in the revised manuscript to temper our conclusions. 

How did individual firms respond to the BP accident, and whether this mitigated the spillover effect on their stock market returns? Regarding responses of other oil and gas firms, not much was done. We have noted this in the manuscript. 

BP’s response took place at multiple levels. In terms of public relations, it issued a series of public apologies. Its so-called “apology commercial”, featuring its CEO Tony Hayward, was criticized because it tended to highlight what BP had done in the past instead of sufficiently and honestly taking responsibility for the oil spill. Hayward’s Congressional testimony was more contrite, but Hayward seemed not to provide clear answers in response to questions posed by the Congressional committee. While professing deep distress, Shell defended offshore drilling as necessary to meet global demand and suggested that if it had been in charge, the accident would not have happened. Exxon Mobil and Chevron also followed the same playbook also did nothing significant unilaterally

3. The biggest issue, though, is in the interpretation of the results. You 

argue in the Discussion section that the industry association plays no role but it might be the opposite: all firms might have reacted fast and in a coordinated way, and might thus have pre-empted the problem. This might be one of the reasons why there is no spillover to the rest of the sector. In that case, the industry association would both play a role in pushing towards self-regulation and in protecting the firms. Note that this pre-emption argument is pretty much what the theoretical literature would predict. See for instance Baron and Diermeier in the Journal of Economics and Management Strategy.

Response

This is an excellent point; theoretically, we could have expected an industry-level response to a collective reputation problem, perhaps coordinated by the American Petroleum Institute. And, this response might have been motivated by the desire to preempt new regulation, as Baron and Diermier have argued. 

The industry responded in a limited (ex post) way. They claimed that the lack of an offshore fleet to deal with spills was a major problem. Thus, instead of individual companies creating their individual offshore relief fleet, the big companies created a Marine Well Containment Company (MWCC), and the smaller companies created HWCG. These fleets can be deployed quickly to the accident site to cap the spilling wells and capture the spilled oil until a relief well is drilled. 

Regarding industry-level action, the American Petroleum Institute (API) announced that it had improved its safety standards and made them available to the public to show the standards in place to promote safety.

However, we do not find much evidence of the evolution of any industry-level voluntary program, which would lead to an ex-ante response to mitigate accidents from happening in the first place. Recall the chemical industry launched its self-regulatory program, Responsible Care, in the aftermath of the 1984 Union Carbide’s Bhopal disaster. It outlined best practices that it expected its member firms to follow to prevent chemical disasters from taking place in the future. Yet, no such ex-ante industry-level response emerged in the oil and gas industry in the aftermath of Deepwater. 

Why might this be so? There are several possible reasons. First, this accident did not result in deaths or massive dislocation of communities (as in wildfires or hurricanes). Thus, after the initial shock and the graphic images of oil pollution and the destruction of marine life, the media tended to move on. Indeed, within six months of the accident, Louisiana (the state most impacted by the spill and highly dependent on oil and gas royalties) politicians were demanding that the federal government not over-regulate this industry. Mary Landrieu, the Democratic senator from Louisiana, demanded that the EPA lifts the ban on BP from securing federal contracts. Interestingly, even the UK government lobbied the Obama administration that BP should not be forced to pay excessive compensation!

Moreover, environmental issues have become deeply partisan. As environmental groups saw the Deepwater accident as an opportunity to push the climate agenda by demanding an end to offshore drilling, partisan identities flared up. The conservatives rushed to defend the offshore oil industry -- and during Trump’s Presidency, even some of the modest new regulations were rolled back. Thus, bipartisan efforts to hold the oil industry accountable were few and weak. A new federal regulation (based on the recommendations of the US Chemical Safety Board), the 2016 Wells Control Rule, emerged. The federal government created a new regulatory body, the Bureau of Safety and Environmental Enforcement (located in the Department of Interior), which is tasked with frequently inspecting offshore facilities for regulatory compliance. However, as Republicans took over the House in 2011, it was clear that new stringent federal regulations aimed at this industry would be difficult to enact (this is probably also why the industry did not feel the need to invest in regulatory pre-emption via voluntary programs). We include this question in the concluding section of the manuscript.

4. Another question is that there are key players that are currently not 

accounted for in your paper: environmental activists and the media. These typically play an important role in creating saliency regarding the issue, thus impacting firms’ reputations. On this, see for instance the article by Breitinger and Bonardi on reputational damages in Business and Politics. There might be heterogeneity regarding environmental issues that way. Of course, one could argue that for a significant issue such as the one you are looking at here, this might be less relevant (since the issue is going to be very salient in any case). But I think you should at least mention this.

Response

Great suggestion. We have noted this in the revised paper. Our sense is that the scale of the disaster was huge that the issue initially got automatic visibility. Environmental activists did mobilize and launched campaigns to boycott BP. But your point is well taken; for smaller, less visible accidents, reputational damages could increase if environmental groups and the media decide to focus on them in their campaigns. We have noted this in the revised manuscript. 

4. Last point that, I think, should be discussed is the difference between 

reputation and stock prices. At the moment, you treat these as the same thing and you hint that their evolution is driven by similar mechanisms. This would be an overstatement. I think you need to take these differences into account in the discussion of your results.

Responses

Fair point. Arguably brand reputations could affect stock market returns, but the opposite effect is less plausible. Reputations could be influenced by consumers’ product experience and value perceptions, while stock market returns could be affected by profits, new product launches, and market volatility. Factors such as industrial accidents or regulatory scrutiny could affect both. We have noted this in the revised paper.

---

## [Decision Letter · Decision Letter 1]

25 Feb 2022

PONE-D-21-16006R1Penalties for industrial accidents: 

The impact of the Deepwater Horizon Accident on BP’s reputation and stock market returnsPLOS ONE

Dear Dr. Holtmaat,

Thank you for submitting your manuscript to PLOS ONE. After careful consideration, I feel that it has merit but does not fully meet PLOS ONE’s publication criteria as it currently stands. Therefore, I invite you to submit a revised version of the manuscript that addresses the points raised during the review process.

Although the manuscript has been significantly improved in this new version, one the reviewers still show major concerns about the methodology used. The reviewer considers that conclusions are not supported by the results, so the manuscript cannot be accepted until these doubts are not attended.

We look forward to receiving your revised manuscript.

Kind regards,

J E. Trinidad Segovia

Academic Editor

PLOS ONE

Reviewers' comments:

Reviewer's Responses to Questions

**Comments to the Author**

1. If the authors have adequately addressed your comments raised in a previous round of review and you feel that this manuscript is now acceptable for publication, you may indicate that here to bypass the “Comments to the Author” section, enter your conflict of interest statement in the “Confidential to Editor” section, and submit your "Accept" recommendation.

Reviewer #1: All comments have been addressed

Reviewer #2: (No Response)

Reviewer #3: All comments have been addressed

2. Is the manuscript technically sound, and do the data support the conclusions?

Reviewer #1: Yes

Reviewer #2: No

Reviewer #3: Yes

3. Has the statistical analysis been performed appropriately and rigorously? 

Reviewer #1: Yes

Reviewer #2: No

Reviewer #3: Yes

4. Have the authors made all data underlying the findings in their manuscript fully available?

Reviewer #1: (No Response)

Reviewer #2: No

Reviewer #3: Yes

5. Is the manuscript presented in an intelligible fashion and written in standard English?

Reviewer #1: Yes

Reviewer #2: Yes

Reviewer #3: Yes

6. Review Comments to the Author

Reviewer #1: Thanks for your great work! You addressed my concern and appreciate your efforts. One last thing. BP's stock prices indeed dropped and suffered at least for a month in the wake of the crisis. If you look at the BP's SP it was $59 on Apr 16, 2010 and dropped to $28 in June. Just two months, but it is clearly true that the SP dropped A LOT. My sense is you should be a little more careful when you argue that the SP was not influenced by the spill; it was affected although in a very short period of time. In the mid- long-term, the SP was recovered from the crisis. So please modify your argument in a more reasonable way. The reputation was damaged, but the SP was not at least in the mid- or long-term "although it had suffered for a very short period time". Please add this one in your abstract. In your current abstract you said "Yet, in terms of financial market returns, we do not find a decline in the stock market returns either in the short term (1-2 years) or the long term (2-7 years)" some people may think short-term should be 1 month, not 1 year. So to avoid any controversy please add that the SP dropped for a month but it was quickly recovered. Thanks.

Reviewer #2: Penalties for industrial accidents: The impact of the Deepwater Horizon accident on BP’s reputation and stock market returns

2/19/2022

Second review

I appreciate the effort done trying to address the methodology issues mentioned in my first review.

I disagree with your interpretation/conclusion there is no evidence that the spill diminished BP stock returns. Your discussion dismisses the fact that -1.458 x 12 = a loss of 17.5% per year and investors did suffer a cumulative loss of about 35% in the 2 years following the accident. That may be statistically insignificant yet is economically significant. Your analysis fails to show the relative underperformance of BP compared to other firms in the oil & gas industry. While such underperformance may not be statistically significant, the paper seems to ignore economically important drops in BP value, especially when compared to its peers.

I still have several concerns regarding the methodologies used in this paper, as outlined below

I recommend authors refer to Kothari, S.P., and Jerold B. Warner, 2004, "Econometrics of Event Studies” and use long term event study methodology. You will probably need a colleague in the finance department to review the manuscript and help you with this paper.

Comments about change of focus

You completely changed your emphasis to stock market returns (title and most discussions) but you probably overdid it. Some examples:

• Rows 103-104 say ”stock market returns provide a measure of the expected future earnings of the firm”. Previously you had stock prices. Stock prices do that, but not stock returns.

• Row 105 says “penalty on stock market returns”. This doesn’t sound right.

• Row 303 “influence stock market returns, especially its future earnings.” Should only say “influence future stock returns”

Please review the usage of “stock price”, “stock returns” and try to make clarifications as needed. Not all instances must be “returns”

Reputation study

Getting a synthetic control is a good idea. However, assigning 76% weight on another oil and gas company raises concerns about the assumptions made and the soundness of the weighting mechanism.

Allowing for the possibility of a contagion effect, you should not be using any oil and gas firms in the synthetic control. That would be a “clean” control. That would remove the issue of using too much of an oil and gas company in your market control. You already remove oil and gas firms when you create synthetic controls for other oil and gas firms (rows 267-268), why not do it for the main subject of study, BP?

In addition to the clean market control, you should create another synthetic with oil and gas firms/brands without BP. That would be your industry control, which would bring these changes:

1. Figure 1 would have a third line (the industry) and contagion can be expected immediately after the explosion. In fact, one can argue that the blue line has the lowest reputation score post-accident right after the explosion, suggesting a minor contagion effect.

2. Figure 2 would have two lines, one for the industry control and one for BP. This chart would show when the industry recovers from any possible contagion.

Figures 3 & 4 makes sense for oil and gas companies/brands. I suggest you remove figure 3.

Stock market study

There is no need to use “the same synthetic control approach described above” (row 322) on returns. In fact, most of the tests used and the charts/tables presented are uncommon in the finance literature. I strongly recommend authors refer to Kothari, S.P., and Jerold B. Warner, 2004, "Econometrics of Event Studies” and use long term event study methodology. If you really want to adapt novel methodologies to stock prices/returns, please find literature that has done it before and include it in reference list. If unavailable, state that you are trying something new, as that may be one of the contributions of the paper. Unfortunately, in my opinion, it is not working.

I am including suggestions below for the figures/tables/methods in the section of Stock Market returns. Hope they help.

In finance we use two main methods to weight stocks into portfolios: market value weight or equal weight. Your method is unclear and unnecessary. For example, how can your calculations assign 40% of the weight to Kellog, the food company? And why do you accept this? Similar issue when you accept 76% of your control to be Shell, an oil company.

Just selecting the 188 firms that are “similar” to BP and assigning equal weight would be enough. If you want to use market value weights, that is also ok. But using your arbitrary weighting mechanism doesn’t look right. In fact, you could even use a broad index as benchmark and that would also work. Personally, I would remove all the oil and gas firms from the 188 firms you matched to BP. I also suggest you study an industry index, to test for a possible contagion effect and as a second benchmark for BP.

Thank you for the appendix. The list of firms and their weights confirmed my suspicion on the methodology. If you follow my suggestions about the formation of portfolios, you won’t need to include it in the final version of the paper.

Table 2 predictor balance is not necessary. What is normally reported in finance studies is the simple (non-normalized) averages of the variables used to match the firm under study. Pre and Post averages can be included.

Figure 5 Total returns of BP vs synthetic control

Don’t call it synthetic control unit when you are analyzing stocks. Common practice in finance is to form/use control portfolios. Why is the starting point Jul 2017? You should make March 2010 your month 0 on any of your charts (figure 6, 7, etc). Chart looks good, but it needs label on the Y axis and change labels: BP instead of treated unit and control portfolio instead of “synthetic control unit”; and make March 2010 equals 100. Probably helpful if you can use the same colors as figure 1. BTW, try to use markers in addition to colors so color blind people can identify which one is BP and which one is the control.

Figure 6, table 3 and the robustness test shown in rows 362-393 unnecessary. Please remove them. There is no reason to test the soundness of the portfolio.

Figure 7 Doesn’t make sense to plot expected and realized. Try just plotting abnormal returns (the difference of the 2). Personally, I would plot the absolute value of the abnormal return or the square of the abnormal return. That way you can see if volatility went up after the event and when volatility goes back to the pre-event levels

Table 4. The regression implied in Table 4 is unclear. Is Y monthly abnormal returns? For explanatory variables, did you use 2 dummies just separating the first 2 years and then the last 5 years? You don’t need a regression for that!. Just t-tests would suffice and would make it clear what is the Pre-event average (constant). My guess is that the -1.458 is the monthly average from Apr 2010 to Mar 2012. However, in another table you say that you start measuring returns May 2010. Why? If the explosion occurred Apr 20, 2010, then 3/31/2010 is the last monthly price pre-event and 4/30/2010 is the first monthly price post event. Obviously, the stock return for Apr 2010 includes the initial reaction to the tragedy. So the return for that month can’t be ignored; it must be included as part of the Post event sample.

Figure 8 Good chart but March 2010 must be the starting point, not Jul 2017. I also suggest using markers so it’s easy to distinguish companies for color blind people.

Figure 9 Total returns. What do you mean total returns? Why do they start at 100? Are they just cumulative unadjusted returns? Starting point must be March 2010. I also suggest using markers so it’s easy to distinguish companies for color blind people. Should also include the industry control portfolio

Table 5 RMSPE ratios for stocks are hard to interpret. The Pre/post ratios for BP are 15, 23 and 87 with expected value of 1 and you still find it insignificant? The calculation of p-value using firm’s rank lacks details and why wasn’t that method used for reputation analysis (in addition to figure 3)?

Including results for other companies is good but show average abnormal returns, not RMSPE ratios. You should also study standard deviation of abnormal returns and include it in table if you find something interesting.

Reviewer #3: Thank you very much for your revision of the paper. I think you have done an excellent job there, and the robustness checks and alternative specifications you provide really make the paper stronger.

I don’t have anything of substance to add.

Just one minor comment: I think you should check the paper for writing and grammatical errors. There are few of them, starting with the first sentence of the Abstract.

The Abstract would also gain by being streamlined. There are repetitions there that could be avoided.

7. PLOS authors have the option to publish the peer review history of their article (what does this mean?). If published, this will include your full peer review and any attached files.

Reviewer #1: No

Reviewer #2: No

Reviewer #3: No

---

## [Author Response · Author response to Decision Letter 1]

29 Apr 2022

Rebuttal Memo

Penalties for industrial accidents: The impact of the Deepwater Horizon accident on BP’s reputation and stock market returns

(PONE-D-21-16006R1)

Dear Reviewers:

Thank you for your comments on our manuscript. We enclose a memo detailing your suggestions (in italics) and outlining how we address them in the revised manuscript. 

Regards,

Authors

Reviewer #1 

1. Thanks for your great work! You addressed my concern and appreciate your efforts. One last thing. BP's stock prices indeed dropped and suffered at least for a month in the wake of the crisis. If you look at the BP's SP it was $59 on Apr 16, 2010 and dropped to $28 in June. Just two months, but it is clearly true that the SP dropped A LOT. My sense is you should be a little more careful when you argue that the SP was not influenced by the spill; it was affected although in a very short period of time. In the mid- long-term, the SP was recovered from the crisis. So please modify your argument in a more reasonable way. The reputation was damaged, but the SP was not at least in the mid- or long-term "although it had suffered for a very short period time". Please add this one in your abstract. In your current abstract you said "Yet, in terms of financial market returns, we do not find a decline in the stock market returns either in the short term (1-2 years) or the long term (2-7 years)" some people may think short-term should be 1 month, not 1 year. So to avoid any controversy please add that the SP dropped for a month but it was quickly recovered. Thanks.

Response

Excellent point. We have modified the abstract to the following: “Yet, in terms of financial market returns, though the stock price dropped drastically in the first two months, we do not find a statistically significant decline in the stock market returns either in the mid-term (1-2 years) or the long term (2-7 years).”

Reviewer #3 

1. Thank you very much for your revision of the paper. I think you have done an excellent job there, and the robustness checks and alternative specifications you provide really make the paper stronger. I don’t have anything of substance to add. Just one minor comment: I think you should check the paper for writing and grammatical errors. There are few of them, starting with the first sentence of the Abstract.

The Abstract would also gain by being streamlined. There are repetitions there that could be avoided.

Response

Thank you. We have checked the paper for errors and streamlined the abstract.

Reviewer #2 

1. I appreciate the effort done trying to address the methodology issues mentioned in my first review. I disagree with your interpretation/conclusion there is no evidence that the spill diminished BP stock returns. Your discussion dismisses the fact that -1.458 x 12 = a loss of 17.5% per year and investors did suffer a cumulative loss of about 35% in the 2 years following the accident. That may be statistically insignificant yet is economically significant. 

Response

We added the following sentences to the manuscript: “Though the decline in stock returns in the wake of the disaster is not statistically significant, we recognize that some might consider it to be economically significant (about 27% decline over a two-year period).”

2. Your analysis fails to show the relative underperformance of BP compared to other firms in the oil & gas industry. While such underperformance may not be statistically significant, the paper seems to ignore economically important drops in BP value, especially when compared to its peers.

Response

Please see the response above. To acknowledge this, we also added the following sentences: “The initial drop however is not recovered over time, and still signals an economically important drop in BP’s value.” And “It also shows how BP’s performance may have dropped vis-à-vis other oil and gas firms, despite the effect of the oil spill not being statistically significant.”

3. I still have several concerns regarding the methodologies used in this paper, as outlined below. I recommend authors refer to Kothari, S.P., and Jerold B. Warner, 2004, "Econometrics of Event Studies” and use long term event study methodology. You will probably need a colleague in the finance department to review the manuscript and help you with this paper.

Response

Thank you for this useful reference. Though we believe the synthetic control method is an improvement on existing event studies, we still use the suggested event studies as a robustness check. More specifically, to address the concerns, we also added robustness checks, using an event study with CAPM to compare BP to the S&P500 minus oil and gas firms and one with only oil and gas firms. Our key results hold.

 

4. Comments about change of focus

You completely changed your emphasis to stock market returns (title and 

most discussions) but you probably overdid it. Some examples:

• Rows 103-104 say ”stock market returns provide a measure of the expected future earnings of the firm”. Previously you had stock prices. Stock prices do that, but not stock returns.

• Row 105 says “penalty on stock market returns”. This doesn’t sound right.

• Row 303 “influence stock market returns, especially its future earnings.” Should only say “influence future stock returns”

Please review the usage of “stock price”, “stock returns” and try to make clarifications as needed. Not all instances must be “returns”

Response

Thank you; we have made changes accordingly.

5. Reputation study

Getting a synthetic control is a good idea. However, assigning 76% weight on another oil and gas company raises concerns about the assumptions made and the soundness of the weighting mechanism.

Allowing for the possibility of a contagion effect, you should not be using any oil and gas firms in the synthetic control. That would be a “clean” control. That would remove the issue of using too much of an oil and gas company in your market control. You already remove oil and gas firms when you create synthetic controls for other oil and gas firms (rows 267-268), why not do it for the main subject of study, BP?

Response

We do it for the other oil and gas firms so we can specifically check for spillover effects. In the main study we don’t do it, because there is a necessary tradeoff here; restricting the set of possible components for the synthetic control may help address concerns about contagion but will also reduce the fit of the model. In the appendix of the revised manuscript, we have added the figure with the synthetic control without oil and gas companies and one with only oil and gas companies. Excluding oil and gas firms from the synthetic control yields essentially the same result as our original model. We have noted this in the revised paper and included the results in the Appendix. 

6. In addition to the clean market control, you should create another synthetic 

with oil and gas firms/brands without BP. That would be your industry control, which would bring these changes:

1. Figure 1 would have a third line (the industry) and contagion can be expected immediately after the explosion. In fact, one can argue that the blue line has the lowest reputation score post-accident right after the explosion, suggesting a minor contagion effect.

2. Figure 2 would have two lines, one for the industry control and one for BP. This chart would show when the industry recovers from any possible contagion.

Figures 3 & 4 makes sense for oil and gas companies/brands. I suggest you remove figure 3.

Response

- Following your suggestion, we added a graph with the synthetic control based on all firms in the data set, all firms minus integrated oil and gas companies and only oil and gas companies to the appendix, both for the reputation analysis and the stock market analysis. We did the same for the CAPM analysis but plotted the lines in different graphs for clarity. 

We hope that by showing similar results when using your suggested methods, we have responded to your input.

Figure 3 is necessary to address concerns that the differences apparent in Figures 1 and 2 are due to simultaneous changes in the reputations of the components of BP’s synthetic control. We acknowledge that contagion effects are an important concern, but we believe that issue has been adequately addressed in the analysis summarized in Figure 4. 

7. Stock market study

There is no need to use “the same synthetic control approach described above” (row 322) on returns. In fact, most of the tests used and the charts/tables presented are uncommon in the finance literature. I strongly recommend authors refer to Kothari, S.P., and Jerold B. Warner, 2004, "Econometrics of Event Studies” and use long term event study methodology. 

 If you really want to adapt novel methodologies to stock prices/returns, please find literature that has done it before and include it in reference list. If unavailable, state that you are trying something new, as that may be one of the contributions of the paper. Unfortunately, in my opinion, it is not working.

Response

We added the following text to explain the synthetic control method vis-à-vis the event study method: “With this we follow others, like Acemoglu et al (31), who have applied the synthetic control method to model the counterfactual in the fields of economics and finance (26,31–37). The synthetic control method is a novel approach in finance that addresses some shortcomings of the more traditional event study method (31,37). The synthetic control method is a variation on other event study methods widely used in finance. The main difference is that the synthetic control method applies specific weights for each unit based on similarities in variables that are driving the outcome variable of interest, making sure the counterfactual is as similar as possible to the treated unit (31). This data-driven creation of a counterfactual is thus a more nuanced approach, than simply using all other companies in the market as a counterfactual. An added advantage is that for making causal inference, weaker assumptions are needed than in the traditional event study methods (38).”

8. I am including suggestions below for the figures/tables/methods in the section of Stock Market returns. Hope they help.

In finance we use two main methods to weight stocks into portfolios: market value weight or equal weight. Your method is unclear and unnecessary. For example, how can your calculations assign 40% of the weight to Kellog, the food company? And why do you accept this? Similar issue when you accept 76% of your control to be Shell, an oil company.

Response

Thank you. Just to be clear, the 40% weight for Kellog and 76% weight for Shell are for the reputation analysis. The synthetic control of stock market returns is based on 187 firms in our dataset, each taking up an almost equal share as you can see in the appendix. To be safe we run the same analysis using CAPM where we compare BP to the overall S&P500 (and S&P500 minus oil and gas and only oil and gas companies in the appendix).

In the main analysis, we are following the well-established synthetic control method (see Abadie A, Diamond A, Hainmueller J. “Comparative politics and the synthetic control method”, American Journal of Political Science. 2015; 59(2):495–510 with 4463 citations), which chooses weights for companies based on similarities in the underlying variables that drive the estimated variable. The selection of components and weight is thus based on similarity. Importantly, we make no assumptions about which brands should be better comparators for BP. We have addressed the concerns about spillovers in our robustness checks.

9. Just selecting the 188 firms that are “similar” to BP and assigning equal 

weight would be enough. If you want to use market value weights, that is also ok. But using your arbitrary weighting mechanism doesn’t look right. In fact, you could even use a broad index as benchmark and that would also work. Personally, I would remove all the oil and gas firms from the 188 firms you matched to BP. I also suggest you study an industry index, to test for a possible contagion effect and as a second benchmark for BP.

Response

Thank you: this is what we do in the CAPM analysis. We compare BP to the S&P 500. In the appendix, we now provide an analysis comparing BP to the rest of the S&P500 firms minus oil and gas companies and BP versus the oil and gas industry. We also assess the abnormal returns compared to the rest of the market.

10. Thank you for the appendix. The list of firms and their weights confirmed my 

suspicion on the methodology. If you follow my suggestions about the formation of portfolios, you won’t need to include it in the final version of the paper.

Response

Since the synthetic control approach is central to our paper, we are retaining the information on the list of firms and the weights. 

11. Table 2 predictor balance is not necessary. What is normally reported in 

finance studies is the simple (non-normalized) averages of the variables used to match the firm under study. Pre and Post averages can be included.

Response

Following the literature, including the predictor balance provides a good sense of how well the counterfactual matches BP on the underlying characteristics. This is quite similar to what is normally reported in finance studies – as you describe. Hence, we retained it in the paper.

12. Figure 5 Total returns of BP vs synthetic control

Don’t call it synthetic control unit when you are analyzing stocks. Common practice in finance is to form/use control portfolios. 

Response

The synthetic control method is comparable to a control portfolio. The companies selected for the control portfolio, when using the synthetic control method, are those that are most similar to BP in the underlying characteristics that drive the stock market returns. We have noted this point in the revised paper.

13. Why is the starting point Jul 2017? You should make March 2010 your 

month on any of your charts (figure 6, 7, etc). 

Response

February 2007 is the starting point of our data. We have revised the paper accordingly. 

14. Chart looks good, but it needs label on the Y axis and change labels: BP 

instead of treated unit and control portfolio instead of “synthetic control unit”; and make March 2010 equals 100. Probably helpful if you can use the same colors as figure 1. BTW, try to use markers in addition to colors so color blind people can identify which one is BP and which one is the control.

Response

Done. 

15. Figure 6, table 3 and the robustness test shown in rows 362-393 unnecessary. Please remove them. There is no reason to test the soundness of the portfolio.

Response

These robustness checks are standard in the synthetic control literature. Here we check how likely it is to receive this discrepancy between BP and the synthetic control, whether this could have occurred if in fact the oil spill had no impact on the stock price. This is akin to the p-value. However, if the editor advises, we can move them to the appendix.

16. Figure 7 Doesn’t make sense to plot expected and realized. Try just plotting 

abnormal returns (the difference of the 2). Personally, I would plot the absolute value of the abnormal return or the square of the abnormal return. That way you can see if volatility went up after the event and when volatility goes back to the pre-event levels.

Response

Done, we now added a plot of BP’s abnormal returns. 

17. Table 4. The regression implied in Table 4 is unclear. Is Y monthly abnormal 

returns? For explanatory variables, did you use 2 dummies just separating the first 2 years and then the last 5 years? You don’t need a regression for that!. Just t-tests would suffice and would make it clear what is the Pre-event average (constant). My guess is that the -1.458 is the monthly average from Apr 2010 to Mar 2012. However, in another table you say that you start measuring returns May 2010. Why? If the explosion occurred Apr 20, 2010, then 3/31/2010 is the last monthly price pre-event and 4/30/2010 is the first monthly price post event. Obviously, the stock return for Apr 2010 includes the initial reaction to the tragedy. So the return for that month can’t be ignored; it must be included as part of the Post event sample.

Response

We measure from May 2010 as we have monthly data from the 1st of each month. So the 1st of April is the last observation before the spill and the 1st of May is the first observation after the oil spill. 

18. Figure 8 Good chart but March 2010 must be the starting point, not Jul 2017. 

I also suggest using markers so it’s easy to distinguish companies for color blind people.

Response

We have updated Figure 8. 

19. Figure 9 Total returns. What do you mean total returns?

Response 

We mean stock market returns.

20. Why do they start at 100? 

Response

Our objective is to create a baseline and make the price fluctuations comparable with one another. 

21. Should also include the industry control portfolio. 

Response

We provide this in the appendix. 

22. Table 5 RMSPE ratios for stocks are hard to interpret. The Pre/post ratios 

for BP are 15, 23 and 87 with expected value of 1 and you still find it insignificant? The calculation of p-value using firm’s rank lacks details and why wasn’t that method used for reputation analysis (in addition to figure 3)?

Response

We explained the calculation of the p-value: “Cunningham (2021) suggests computing the Post/Pre RMSPE ratio for the treated unit and each component of the synthetic control and seeing where the treated unit ranks in that distribution (39). Table 5 shows BP’s ranking in the distribution of RMSPE ratios. We calculated this ranking for our full sample period, the two years following the disaster (“Short-term”), and the period two to ten years after the disaster (“Long-term”)” “The p-value is simply the rank divided by the total number of firms, which is 188 in BP’s case.” Figure 3 is a visual representation of the RMSPE ratios, showing that BP’s ratio stands out vis-à-vis the other oil and gas firms, making a table with RMSPE ratios and p-values redundant. Given that the financial synthetic control consists out of many more component firms, we use p-values to give an impression of the relative size of RMSPE-ratios, instead of a figure.

23. Including results for other companies is good but show average abnormal 

returns, not RMSPE ratios. You should also study standard deviation of abnormal returns and include it in table if you find something interesting.

Response

RMSPE ratios are standard in the synthetic control literature and are used here to determine the statistical significance of the results we observed. The cumulative abnormal returns of other oil and gas companies are captured in figure 9. The standard deviation of abnormal returns is interesting and important, but not directly related to our research question.

---

## [Editor Report · Decision Letter 2]

9 May 2022

Penalties for industrial accidents: 

The impact of the Deepwater Horizon Accident on BP’s reputation and stock market returns

PONE-D-21-16006R2

Dear Dr. Holtmaat,

We’re pleased to inform you that your manuscript has been judged scientifically suitable for publication and will be formally accepted for publication once it meets all outstanding technical requirements.

Kind regards,

J E. Trinidad Segovia

Section Editor

PLOS ONE
---

## [Editor Report · Acceptance letter]

23 May 2022

PONE-D-21-16006R2 

Penalties for industrial accidents:
The impact of the Deepwater Horizon accident on BP’s reputation and stock market returns 

Dear Dr. Holtmaat:

I'm pleased to inform you that your manuscript has been deemed suitable for publication in PLOS ONE. Congratulations! Your manuscript is now with our production department. 

Kind regards, 

on behalf of

Dr. J E. Trinidad Segovia 

Section Editor

PLOS ONE